

# A Pathways Analysis Dashboard prototype for multi-risk systems

Julius Schlumberger[1,2], Robert Šakić Trogrlić[3], Jeroen C.J.H. Aerts[1,2], Jung-Hee Hyun[3],
Stefan Hochrainer-Stigler[3], Marleen de Ruiter[2], and Marjolijn Haasnoot[2,4]

[1] Deltares, Boussinesqweg 1, 2629 HV, Delft, The Netherlands
[2] Vrije Universiteit Amsterdam (VU) Instituut voor Milieuvraagstukken (IVM), De Boelelaan 1111, 1081 HV, Amsterdam, The Netherlands
[3] International Institute for Applied Systems Analysis (IIASA), Schlossplatz 1, Laxenburg, Austria
[4] Utrecht University (UU) Faculty of Geosciences, Princetonlaan 8a, 3584 CB Utrecht, The Netherlands

**Correspondence:** Julius Schlumberger (julius.schlumberger@deltares.nl)

**Abstract.** With accelerating climate change, impacts will compound and cascade, making them more complex to assess and manage. At the same time, tools that help decision makers choose between different management options are very limited. This study introduces a visual analytics dashboard prototype designed to support pathways analysis for multi-risk Disaster Risk Management (DRM). Developed through a systematic design approach, the dashboard employs interactive visualisations

of pathways and their evaluation - including Decision Trees, Parallel Coordinates Plots, Stacked Bar Charts, Heatmaps, and Pathways Maps - to facilitate complex, multi-criteria decision-making under uncertainty. We demonstrate the utility of the dashboard through an evaluation with 54 participants at varying levels and disciplines of expertise. Depending on the expertise (non-experts, adaptation / DRM experts, pathways experts), users were able to interpret the options of the pathways, the performance of the pathways, the timing of the decisions and perform a system analysis that accounts for interactions between

the sectoral DRM pathways with precision between 71% and 80%. Participants particularly valued the dashboard's interactivity, allowing for scenario exploration, adding additional information on demand, or offering additional clarifying data. Although the dashboard effectively supports comparative analysis of pathway options, the study highlights the need for additional guidance and onboarding resources to improve accessibility and opportunities to generalise the prototype developed to be applied in different case studies. Tested as a standalone tool, the dashboard may have additional value in participatory analysis and

modelling settings. This study underscores the value of visual analytics for the DRM and Decision Making Under Deep Uncertainty (DMDU) communities, with implications for broader applications across complex and uncertain decision-making scenarios.

## 1 Introduction

Societies face complex disaster risk management (DRM) decisions under uncertain changing conditions influenced by climate

change and socioeconomic factors (Buskop et al., 2024 in review; Simpson et al., 2023; Walker et al., 2008). For example, New York must plan for sea level rise and storm surges while considering adaptive responses such as protection, adaptation, or retreat (Haasnoot et al., 2021). In Australia and the western United States, managing forest fire risk requires navigating uncertainties in forest management, urban planning, and climate projections (Johnson et al., 2023; de Rigo et al., 2013). These



examples illustrate that DRM decisions anticipate evolving risks shaped by the interaction of natural and human systems and
should incorporate a forward-looking approach.

To address these complexities, pathway thinking, particularly within the Decision-Making Under Deep Uncertainty (DMDU) community, has become prevalent. For example, frameworks like Dynamic Adaptive Pathways Planning (DAPP) guide flexible and robust decision making in plausible futures (Haasnoot et al., 2024). Pathways thinking promotes adaptive decision making over time, allowing stakeholders to identify immediate and long-term options, avoid lock-ins, and implement staged risk
reduction measures (Hanger-Kopp et al., 2022; Thaler et al., 2023; Haasnoot et al., 2019; Cradock-Henry and Frame, 2021; Werners et al., 2021).

Recently, DAPP has been adapted for multi-risk settings (DAPP-MR), which consider interactions between different hazards and sectors (Schlumberger et al., 2023). Such interactions can cause cascading impacts between sectors and regions or interaction effects between risk management strategies (de Ruiter et al., 2021; Nilsson; Simpson et al., 2021; Kool et al., 2024).
To manage these interactions, DAPP-MR takes a stepwise approach to find combinations of pathways that are viable for all sectors and a range of risks. This method first analyses sector risk pathways individually before increasing complexity by integrating pathways across multiple sectors and risks and assessing pathway combinations under diverse future scenarios. Despite its promise, evaluating pathways in multi-risk settings remains challenging because of the large number of combinations of pathways, risks, sectors, and future scenarios. A recent case study on DAPP-MR with four actors and two hazards illustrated
the difficulty in analysing such multidimensional data, highlighting the need for better visualisation tools to unravel complexity and support DRM (Schlumberger et al., 2024).

Information visualisation, which facilitates the exploration, sense making, and communication of complex data (Hindalong et al., 2020; Salo and Hämäläinen, 2010), has become a valuable tool for the analysis of pathways. However, visualisations in DMDU often lack justification for design choices or evaluation of their support for decision making (Hadjimichael et al.,
2024). Only a few studies evaluate visualisation tools based on cognitive science principles and user feedback (Bonham et al., 2022; Shavazipour et al., 2021). Visual analytics can help analyse DRM pathways in a multi-risk environment as they enable interactive data exploration, fostering an iterative (Shneiderman, 1996) and collaborative analysis process (Ceneda et al., 2017; Bajracharya et al., 2018). However, visual analytics applications in DMDU remain limited, with few studies demonstrating their effectiveness for DRM (Bonham et al., 2024; Hadka et al., 2015; Woodruff et al., 2013).

In this study, our aim was to design and evaluate a visual analytics dashboard tailored for analysing pathways in multi-risk settings. We develop a set of visualisation alternatives based on a systematic design process (Munzner, 2009) and embed them in an interactive dashboard to support the analysis for a wide range of potential users. The developed dashboard is evaluated through feedback from 54 potential users. The paper is structured as follows. Section 2 describes the design process and evaluation approach; Section 3 presents and discusses the evaluation results; Section 4 provides reflections; and Section 5
concludes.



## 2 Methods

Following a systematic approach (Munzner, 2009), we used a five-step iterative design process (Figure 1) to create an interactive pathways analysis dashboard. The following subsections provide a concise overview of the design process, with further details available in Appendices A to C and Supplementary Material. As we refer to multiple types of steps and questions in the following sections, we want to briefly distinguish between key terms used. In the following, we will use 'design steps' to refer to the procedure of developing and evaluating the dashboard. We use 'themes of analysis' to differentiate between major components of pathways analysis and 'questions of interest' to describe questions that users need answers for. These questions are translated into 'analysis operations' in abstracted terms.

In the first design step, users and key questions for pathways analysis are identified to ensure that visualisations are designed for the right purpose (Hindalong et al., 2020). In the second step, these key questions are translated into analysis operations, abstractions of what essential visualisation characteristics will be used (how) to extract the relevant information from the visualisation, used to answer the key questions (Munzner, 2009). Afterwards, in step three, the raw model output data is transformed into visualisable formats to support analysis operations (Correa et al., 2009; Munzner, 2014). In step four, visualisation types are chosen that align with the transformed data dimensions and analysis operations. Lastly, in step five, user feedback is collected through a survey to assess the objective fit (ability to gain intended insights) and subjective fit (ease of information extraction).

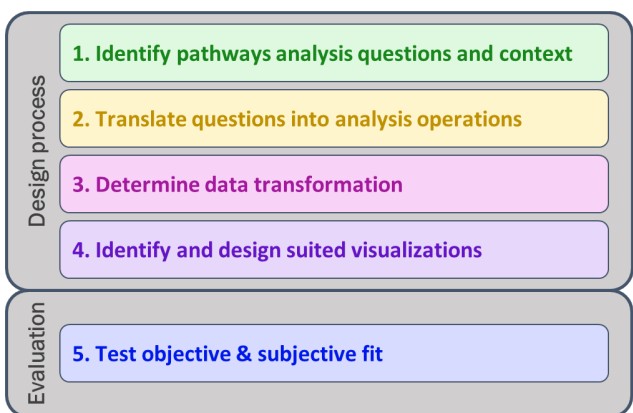

**Figure 1.** Design process to develop a visual analytics dashboard and evaluate its objective and subjective fit.

### 2.1 Identify pathways analysis questions and context

In the first step of the design of the pathways analysis dashboard, we defined the pathways analysis context, including identified target users and their capacities, and formulated key questions of interest. Six semi-structured interviews and two 60-minute focus groups (n = 21) collected feedback on a preliminary pathways analysis framework and potential users, based on previous





studies (Schlumberger et al., 2022, 2024). The interviews and workshops followed the guidelines of Hove and Anda (2005) (Supplementary Material S1 for details).

Feedback from interviews indicated that the pathways analysis process needs to be clearly guided, introducing relevant concepts and the purpose of the analysis, as stakeholders often have limited time and resources. Early adopters involved in pathways analysis come from diverse disciplines and administrative levels, motivated by (i) understanding multi-risk interactions and system-wide effects, (ii) identifying sector-specific low-regret pathways, and (iii) identifying system-wide low-regret pathways combinations. Four themes of analysis emerged with more detailed analysis questions (Table 1): 'What are the pathway options?', 'How do the pathway options perform?', 'How are these pathway options mapped over time?' and 'Which combinations of pathways serve multiple hazards and sectors?' Most of the questions focus on sectoral perspectives, and stakeholders prioritise different indicators, timescales, or scenarios. Therefore, we assume that stakeholders are involved in a broader participatory modelling process to specify analysis criteria to develop forward-looking DRM pathways. Given the systems perspective of multi-risk DRM, the process also involves elements of collaborative learning (Laal and Laal, 2012), such as knowledge exchange and discussion among stakeholders with diverse needs and interests, to develop a cohesive DRM strategy across sectoral boundaries.

## 2.2 Translate questions into analysis operations

In the second step, we abstracted the pathways analysis questions into analysis operations to clarify the analysis goals and methods (Table 1), according to standard design practices (Amar et al., 2005; Wehrend and Lewis, 1990). These abstractions help to clarify why users are engaging in the analysis (e.g., finding trends, outliers, etc.) and which types of analyses they would like to conduct (e.g. compare different alternatives, discover patterns, etc.). (Brehmer and Munzner, 2013). The abstraction identifies which properties of a pathway data set are most relevant and what properties of a visualisation will be used to find answers to the question of interest. The bold terms in Table 1 used for the description of the analysis operations are based on Brehmer and Munzner (2013) (definitions in Table A). Furthermore, we used the term 'candidate' to refer to both a pathway option or a specific action as part of a pathway. Furthermore, we used the term 'attribute' to refer to any property or value of the candidate (such as name, description, performance objective, etc.). We also used the term 'data subset' to express that some datasets to be visualised will be only subsets of the whole dataset, for example, showing values for objective keys for a specific time horizon, scenario or combination of pathways.

## 2.3 Determine data transformations

After defining user analysis needs, this step focused on suitable data transformations to visualise these needs. DRM data for pathway analysis are multidimensional, spanning scenarios with both external (climate, socio-economic) and internal (actor measures) uncertainties. However, effective visualisation typically handles up to five dimensions to maintain clarity (Mackinlay, 1986; Siirtola, 2007). This means that choices have to be made regarding how to reduce dimensionality and the number of data points shown. Both depend on the interest of the stakeholders and their previous experience or analysis capabilities (Bonham et al., 2024; Kwakkel et al., 2016).



**Table 1.** Four themes of pathways analysis (first column), related questions of interest (second column) and corresponding analysis operations (third columns). *Italic* terms in the second column mark analysis operations which are defined in the Table A1.

| Theme | Question of Interest | Analysis operation |
|---|---|---|
| **A. What are the pathways options?** | What measures are available for addressing the identified risk? | *Select* individual candidates to *lookup* different attributes of the candidates. |
| | Which measures are short-term actions or long-term options? | *Arrange* relevant candidates to *identify* the distribution of candidates. |
| | How do pathways options differ? | *Select* candidates to lookup and *compare* attributes of the candidates. |
| **B. How do the pathways options perform?** | How does each pathway perform across key performance criteria? | *Filter* or *select* candidates based on attributes (1) to *compare* trends in attributes across candidates and (2) to *identify* candidates with attribute outliers. |
| | How robust are these pathways under different future scenarios and on different time horizons? | *Change* between different data subsets to *explore* correlation and similarity of candidate attributes across different subsets. |
| | What are synergies or trade-offs between different performance criteria? | *Order* attributes of different candidates to *identify* correlations between attributes. |
| | How does the performance of pathways change when accounting for multi-risk interactions? | *Change* between different data subsets and *overlay* candidate attributes of different subsets (1) to *explore* candidates with attributes of high and low similarity across the data-sets (2) to *locate* the outlier subsets with the strongest similarity/difference of candidate attributes. |
| **C. How do these pathways options map out in time?** | When are points reached where a change in strategy is required? | *Select* candidates to *lookup* attributes (time, name, additional information). |
| | How does the timing of these points change for different future scenarios? | Arrange attributes of candidates to identify the distribution of attributes |
| | | *Change* between different data subsets to *explore* candidates with attributes of high and low similarity across the data-sets. |
| | How do multi-risk interactions affect the timing of these points? | *Change* between different data subsets, *overlay* candidate attributes of different subsets to *explore* the similarity of candidate attributes across the data-sets. |
| **D. Which combinations of pathways serve multiple hazards and sectors?** | How do individual pathway options align or conflict with those of other actors? | *Select* candidates, *overlay* candidate attributes of different data subsets to *identify* trends in similarity across attributes. |
| | | *Change* between different candidates, *overlay* candidate attributes of different subsets to *compare* outliers in similarity across attributes and candidates. |
| | What are synergies and trade-offs of collaborating with other actors? | *Change* between different candidates, *overlay* candidate attributes of different subsets to *compare* outliers in similarity across attributes and candidates. |



In DMDU, statistics-based summary methods are commonly used for dimensionality reduction to calculate the robustness of pathways. Robustness is defined as the ability of a policy option to perform well across an ensemble of uncertainties while minimising regret. Various performance robustness indicators can be calculated using combinations of statistical properties (e.g., mean and standard deviation) of the data set in a (sub)set of scenarios (Bartholomew and Kwakkel, 2020). Furthermore, data density is often reduced by filtering (Brehmer and Munzner, 2013). For example, while the performance of different pathways could be analysed for each year of the planning horizon, specifying (a set of) times of interest reduces the number of relevant data points to be considered for the analysis (e.g., Kwakkel et al., 2015; Schlumberger et al., 2024).

To explore relevant transformations, data from a case study on the Waal River in the Netherlands was used, modelling flood and drought interactions across agriculture, urban, and shipping sectors over a 100-year period with a resolution of 10 days (Haasnoot et al., 2012; Schlumberger et al., 2024). Each sector manages climate risks by implementing sequences of DRM measures referred to as 'DRM pathways'. The pathways of each sector are evaluated based on sectoral objectives in combinations with the DRM pathways of different sectors and accounting for climate variability and climate change scenarios (Schlumberger et al., 2024). Details on the case study and data flow are provided in Supplementary Material S2.

## 2.4 Designing information visualisations to complete the analysis operations

When developing the interactive dashboard and integrating fit-for-purpose visualisations, we focused on two components: 1) designing information visualisations to complete the analysis operations and 2) creating an environment that serves a wide range of users to gain additional insight into the concepts and purpose of the themes of analysis.

The systematic design process resulted in a dashboard environment, which supports users to analyse DRM pathways and their effectiveness to reduce the complexity of climate risk analysis through interactive visualisations. The visualisations on the dashboard are aligned with analysis operations, creating an accessible and interactive environment that serves a wide range of users. Built with Python 3.10, the dashboard uses open source tools (Dash, Plotly, Pathways Generator[1]) and is hosted on Heroku. The URL of the dashboard, www.pathways-analysis-dashboard.net, is accessible with a Web browser and an Internet connection.

### 2.4.1 Designing information visualisations to complete the analysis operations

We identified five visualisation methods as potentially suitable (Figure 2). Each visualisation was deemed suitable to answer all relevant questions of interest (Table 1) per theme of analysis A to D. It was also taken into account whether the visualisations are scalable to be modified and used for the system analysis theme. Visualisations were identified based on a literature review in the field of visualisation research and cognitive studies (e.g., Börner et al., 2019; Munzner, 2014) and the DMDU community (e.g., Gold et al., 2022; Gratzl et al., 2013; Haasnoot et al., 2024; Hindalong et al., 2020; Moallemi et al., 2020; Trindade et al., 2019), author discussions and preliminary testing. More details on the design process and the elaboration of the final visualisation are provided in the Appendix B. We implemented and tested the following visualisation methods per analysis theme:

---

[1]https://github.com/Deltares-research/PathwaysGenerator





*A) Explore the pathways options*: To explore the available pathway options, we use **Decision Trees (DT)**, which display different sequences of choices in a branching structure, helping to understand the hierarchy of decision points (Shneiderman, 1996).

*B) pathway performance*: For the assessment of the performance of the pathways, we identified three alternatives: 1) **Parallel Coordinates Plots (PCP)** show the pathways options as a polyline across multiple objectives on the parallel y-axis, allowing us to observe correlations and trade-offs (Itoh et al., 2017; Siirtola, 2007). 2) **Stacked Bar Charts (SBC)**, highlight how multiple objectives contribute to cumulative performance for different pathways options while distinguishing individual contributions by colour (Gratzl et al., 2013; Hindalong et al., 2020; Streit and Gehlenborg, 2014). 3) **Heatmaps (HM)** provide a straightforward matrix view, using colour to convey quantitative values, facilitating quick comparisons within and between categories (Munzner, 2014; Shavazipour et al., 2021).

*C) Pathway timing*: To visualise decision timing, we identified **Pathways Maps (PM)**, a 'Metro-style map', visualising decisions over time or changing conditions, transfers between decisions, and path dependencies (Haasnoot et al., 2024).

*D) Pathway timing and performance for the entire system*: We used modifications of PCP, SBC, HM, and PM to allow visualisation of the combined effects. For visualisations to analyse the performance, we added an interactive element allowing users to scale the number of combinations to analyse. A similar interactive element was added for the pathways to explore the interaction of specific combinations of pathways. The colour schemes were additionally adjusted to accommodate the increased number of objectives/sectors.

### 2.4.2 Creating an environment that serves a wide range of users

The dashboard's multi-page layout separates the four themes of analysis, guiding users through a stepwise analysis. Users first analyse their specific sectoral pathway options, then their pathway performance, and finally the timing of adaptation tipping points to identify a short list of promising pathways that best meet their specific objectives. In the last step of analysis, different sectoral actors bring their individual shortlisted pathways to one table to explore the combination effects of different sectoral pathways in a collaborative learning step. The general dashboard structure is shown in Figure 3, and possible options to modify the visualisation are available, e.g. selecting a certain time horizon or climate scenario, or choosing different robustness definitions to determine the performance robustness. Additionally, guidance on how to read the visualisation is provided, and explanations for key terms relevant to the pathways analysis (e.g. robustness, scenario) can be obtained on demand.

### 2.5 Test objective and subjective fit

To evaluate the effectiveness of the dashboard and the visualisations, we embedded a 15-30 minute questionnaire based on best practices (Kosara et al., 2003; Conati et al., 2014; Dimara et al., 2018). The set of survey questions can be found in the Appendix C. The survey involved a broad and diverse range of 54 potential users and experts in the fields of information visualisation, disaster risk management, and pathways thinking and beyond. Responses were screened out for validity, notably excluding dummy inputs (e.g. combination of no free-text feedback, identical Likert-scale evaluations, overarchingly random inputs) and duplicates (which happened if they kept their sessions open too long). Although participants were encouraged





**Figure 2.** Set of visualisation types for pathways analysis. visualisation of pathways options by means of DT (a). Interactive components offer information on demand. SBC (b), PCP (c) and HM (d) to explore pathways performance across multiple objectives for different pathways options. Pathways map to investigate the timing of decision-making (e).

to complete the entire questionnaire, intermediate results were saved per analysis theme. For the evaluation, we consider all available data, even if the participants did not complete the entire questionnaire.

The objective fit of the dashboard and its visualisations was evaluated by the precision of the responses to a set of analysis questions compared to the answers that the authors deemed correct (Gratzl et al., 2013). We chose a varying set of simple



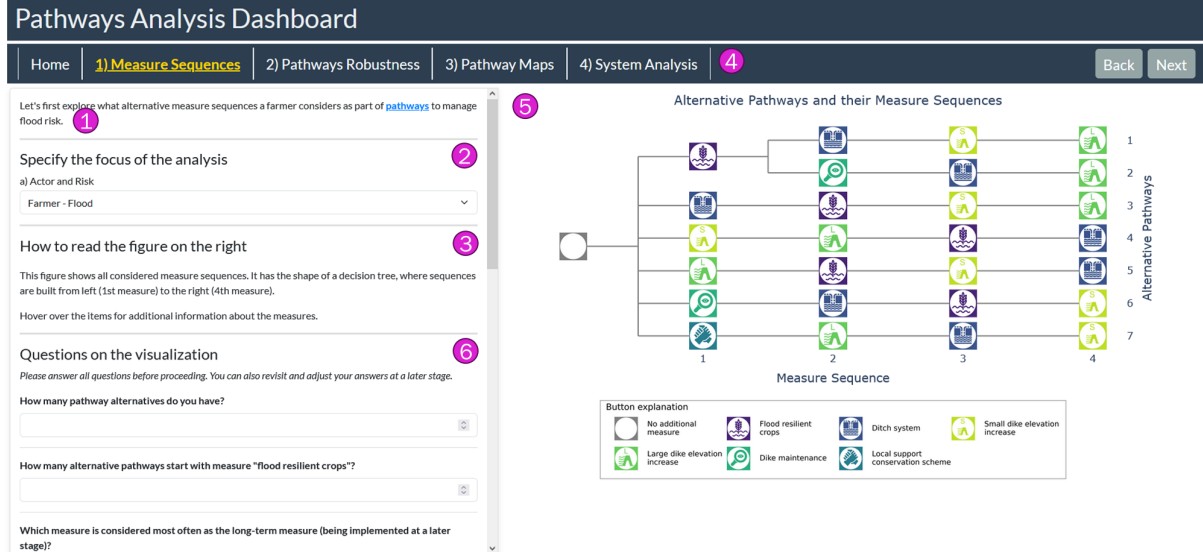

**Figure 3.** Dashboard outline. To offer guidance and flexibility, it contains the following elements: A short description of the analysis task at hand, including additional information on demand regarding key concepts (1). A section to select the relevant analysis focus. This section varies from theme of analysis to theme of analysis (2). A short explanation how to read the visualisation and what options for interaction are given (3). A Navigation bar, offering means to navigate between the different themes but also clarifying the current theme of analysis (4). The interactive visualisation itself, used for the analysis operations (5). The survey was embedded into the dashboard to improve accessibility (6).

and more complicated questions and performed a dashboard analysis from an aggregated level to the specific analysis task of a given visualisation (Plaisant, 2004). In the analysis, we put a stronger emphasis on questions where the precision of the response was below 70% to discuss challenges and misconceptions that were widely represented among survey participants. To evaluate the subjective fit of the dashboard, participants were asked to express their agreement with sentences stating that the visualisation was easy to understand, that they are confident in their response, that they had enough information to effectively use the visualisation and that they would use this type of visualisation for similar questions (Dimara et al., 2018) using the 5-point Likert scale ('totally disagree' to 'totally agree'). Qualitative feedback provided additional anecdotal evidence on dashboard strengths and areas for improvement (Conati et al., 2014). We now move on in presenting the results.

## 3   Evaluating the visual analysis dashboard

We collected feedback from 54 participants, with responses from all participants on visualisation of the pathways options, 85% (n = 46) on the robustness of performance, and 81% (n = 44) on decision timing. Approximately 70% (n = 38) completed the survey for all analysis themes. Most of the participants (78%, n = 42) worked in research, 9% in the private sector (n=5), and 96% did not report visual impairments (n = 52). The expertise of the participants included DMDU / Pathways (n = 13), Climate




Adaptation / DRM (n = 17) and other fields such as Architecture, Computational Science, and Governance (n=24) (see Table C2).

In general, the dashboard provided relevant information to the participants, see Figure 4. The correct answer rates were above 70% for most expert groups and analysis themes, with one outlier for the analysis of the system for non-experts (61%). Expertise particularly influenced success in decision-timing and system analysis, favouring those with prior experience in pathways and system thinking. The subjective fit was similarly expertise dependent, with DMDU experts more likely to find the visualisations clear, be confident in their responses, and foresee using them again, while non-experts were more neutral.

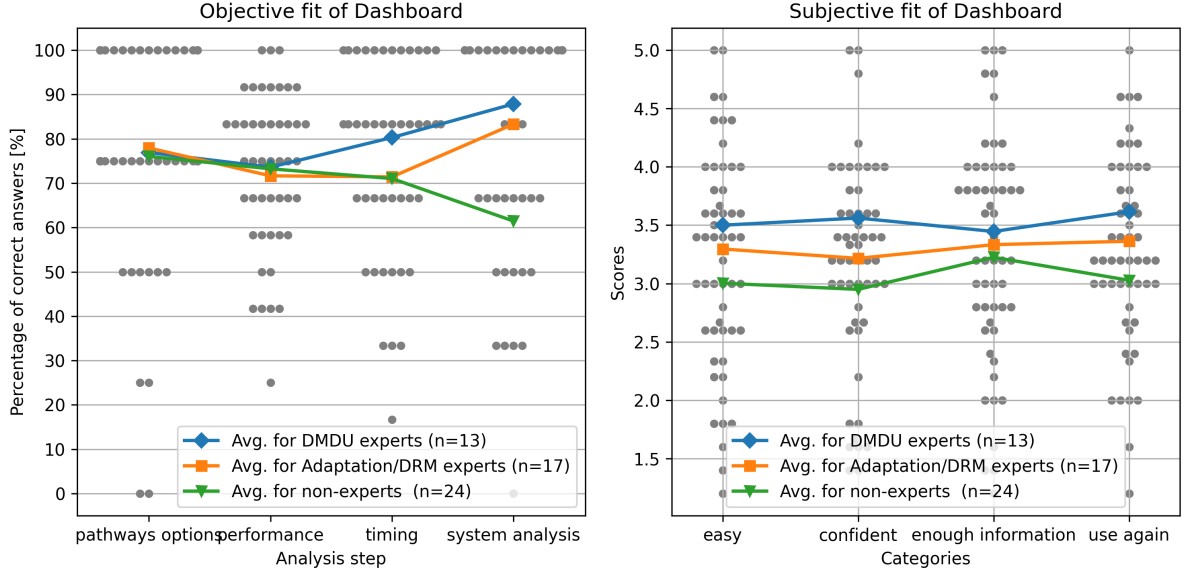

**Figure 4.** Percentage of correct answers (left) and subjective evaluation (right) for all participants (gray dots) and averaged across participants of the same expertise (coloured lines and markers) for the four themes of analysis. Note that the number of participants is different for each step: exploration of pathways options (n=54), pathways performance (n=42), pathways timing (n=40), system analysis (n=35).

### 3.1 Evaluation of the dashboard to support pathways options analysis

Pathways options were analysed using a DT (Figure 2a). The objective fit was assessed using four questions, see Figure 5left, with the participants accurately answering most of the questions. Questions A2 and A3 were less well answered (hit rates: 60%, n = 54) for different reasons. Question A2 required participants to identify the measure that is the starting measure in most of the pathway options. One participant reflected that *'information is spread over the entire figure [...]. I need to read the y-axis on the right and move back to the left.'* Similarly, participants pointed out that the visualisation design did not intuitively lead the focus of a participant from the left to the right (e.g. *'Connecting lines could have arrowheads, would make the sequence visually more intuitive'*).



Regarding Question A3, which required participants to identify the most frequent measure that is being implemented in the long term. The question lacked clarity about the definition of 'long-term' (*'What is most the option to be implemented at a later stage. 'Large dike increase' is the last option most often. However, 'small dike elevations' occurs most often in the last two steps.'*). We considered only the last option to be defined as long-term, but fifteen out of the 19 incorrect answers consider the past two sequence steps as long-term, which could arguably be correct as well.

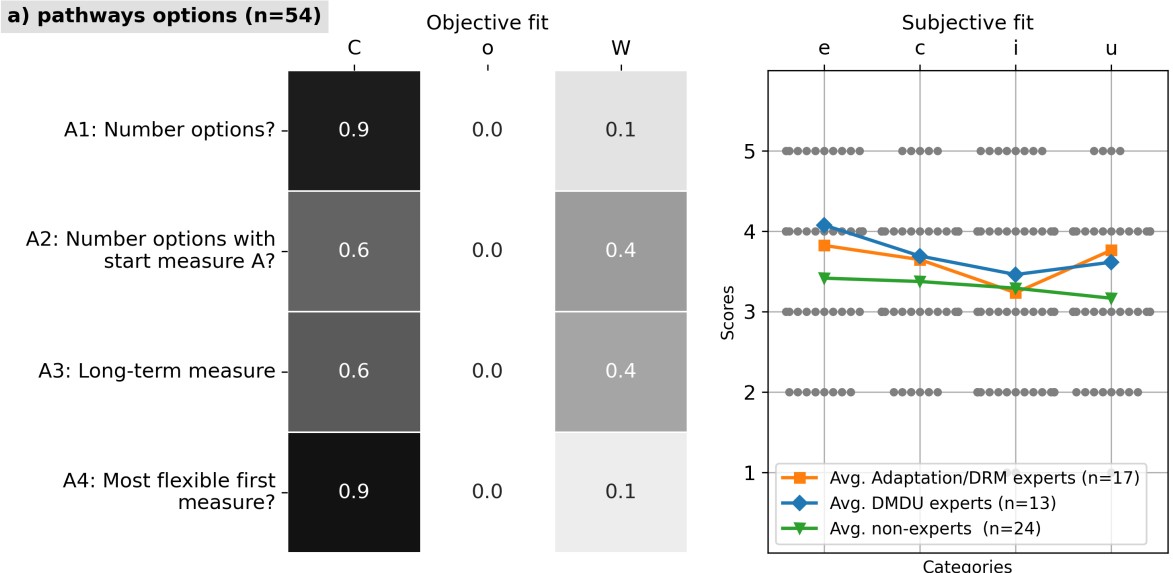

**Figure 5.** Evaluation of the dashboard for the first theme of analysis ('What are the pathways options?') based on the inputs from the users (n=54). Left: Evaluating the objective fit based on the share of correct answers (C) compared to wrong (W) and partially correct answers (o) (for the full set of questions refer to Table C1 in the Appendix). Right evaluating the subjective fit, differentiated in how easy they find the visualisation (e), how confident they are about the made choice (c), if the had enough information (i) and whether they would use this visualisation type for similar problems (u).

The evaluation of the subjective fit is overarchingly positive as summarised in Figure 5right. Participants generally agree that the visualisation provides enough information, is easy to understand, makes them feel confident that they answered correctly, and would be used for similar problems. Subjectively, participants positively valued the colour scheme and symbols (e.g. *'The icons are clear, the colours assist distinguishing the measures'*) but noted issues with colour logic and icon density (e.g. *'There are a lot of symbols, which if you're not used to them takes time to read the figure. Greater difference in colours might be useful.'* or *'colours for measure are not logical (elevation should be brown, crops yellow, ditch blue...)'*). The participants appreciated the interactive nature of the visualisations (e.g. *'I like the interactive nature of the figure. The extra information that comes when you hover over an action is helpful.'*). At the same time, multiple participants criticised the lack of background information that makes it difficult to make sense of the pathways options presented and why some are possible and others are not (e.g. *'no*





*additional information on the feasibility of each pathway, which makes it more difficult to understand why some measures need to be in an earlier stage compared to others or why one is more flexible.'*).

## 3.2 Evaluation of the dashboard to support pathways performance analysis

For the performance analysis, the participants were randomly presented with a PCP, HM, or SBC (Figures 2b to 2d). PCP and SBC outperformed HM in clarity and correctness, as shown in Figure 6. Subjectively, participants found HM challenging
to interpret and would not use such a visualisation for similar problems, whereas PCP were appreciated for dealing with the multi-objective analysis of performance robustness and would use it again despite lower confidence in their answers chosen. The evaluation of the subjective fit is somewhat ambiguous. It should be noted that the DMDU experts evaluated PCP much more compared to the other expert groups, while the patterns were quite similar for SBC. Non-experts were particularly uncertain about their responses when using PCP and HM. Although participants subjectively tended to agree that SBC offered
sufficient information and that they are confident in their responses, they disagreed that the visualisation was easy to use and thus tended not to use it for similar problems.

The participants mentioned some challenges that were relevant for all different visualisations. The participants had a particular struggle to understand the concept of robustness of the pathways and thus how they could deduce information about robustness from the figure (e.g. *'I struggle to understand how to evaluate robustness'*). One participant asked for more infor-
mation on how it is calculated (e.g. *'Black-box how performance robustness was calculated.'*). Similarly, participants referred that they would need more context information to understand why the pathways options are analysed and where the differences come from (e.g. *'I don't understand, but want to know how the strategies were identified and if the differences between them are meaningful.'*) and how terms such as synergies and trade-offs are applied in this context (e.g. *'it is not clear on the difference between synergy loss and trade-off loss [...] Some explanation of how these terms are applied here and are different from each
other in their application to farmer strategies could help.'*). Multiple participants suggested additional guidance (e.g. *'Put a video with a talk to help navigate with an example.'* or *'Everything is useful, but need to put an example first.'*).

For **PCP**, question B3 was not answered correctly by any participant (n= 13), while 50% of participants provided partially correct answers to Question B6. For question B3, the task was to identify the pathway option with the best robustness and required a combined consideration of robustness performance across multiple objectives. In the introductory text, it is men-
tioned that robustness is evaluated across objectives. However, no further details on how to carry out this evaluation between objectives were provided. Additionally, aggregating this performance across parallel axes is a recognised weakness of this type of visualisation (Siirtola, 2007). For question B6, asking to identify the pathway(s) with the best robust performance in relation to one objective when accounting for interactions, it appears that similar colour coding of lines representing different pathways
led participants to incorrect answers (*'difficult to follow the lines across the figure - some colours were difficult to distinguish, so hard to determine what the value was for some of the pathways'*). This also implied that some participants did not use the full potential of the interactive elements, which would have allowed them to filter pathway options that fall in certain ranges along each of the axes. The general feedback was positive (*'I've never seen a figure like this and I actually find it a very good way to*





*summarise key information that I (trying to put myself in shoes of a farmer) would want to see.'*). Participants appreciate how
the figure allows for the comparison of multiple variables simultaneously and visually represents different scenarios, helping to evaluate the efficiency of adaptation investments. The use of colours and multiple axes to show robustness scores is noted as a valuable feature that makes information easier to interpret (*'The different colours and the different axes illustrating the different robustness scores'*).

For **SBC**, more participants were able to correctly answer Question B3 (47%, n = 9), understanding that the robustness across objectives was measured by the shortness of the stacked bar. One participant interpreted the length in the opposite way, selecting the pathway with the longest bar as the most robust pathway. The number of partially right answers to question B3 can be associated with a bug (before fixing the bug: 1 out of 9 participants correct; after fixing bug: 8 out of 10 participants correct) in the early version which resulted in bars of equal performance having different lengths (*'the crop productivity loss
bar looked different for different pathways, but the information shown by hovering was that the loss was same'*). Most of the participants did not correctly answer Question B6 (hit rate 30%), identifying the pathways with the best performance with respect to one objective when considering interactions with another sector. The incorrect answers seem to be misled by the representation of synergy and trade-off effects in the visualisation as additional bars of different length (*'I don't know what the synergy or trade-off effects mean.'*). Multiple participants indicated that they would prefer more information. One participant
stated: *'The sizing of the bars is not 100% intuitive. Potentially adding a x-axis would help.'*
The participants appreciated interactive features such as hovering, which allowed participants to engage with the content and explore various climate scenarios, helping to visualise interactions effectively (*'Very clear descriptions on the bottom when hovering over each box'*). Furthermore, participants confirmed that the colours and shading used in the figure help readability, making complex information more accessible (*'The colours and shading help to understand the graphic'*).


For **HM**, participants particularly struggled with questions B3, B5 and B6 (Figure 6). Regarding Question B3 (hit rate 30%, n=11), an explanation is that the robustness between objectives is not clearly coded in the visualisation of HM (and PCP) compared to SBC. For Questions B5 and B6 (hit rate 40% and 10%), participants were asked to discover patterns of interaction effects. 7 out of 9 incorrect answers indicated that they were unable to discover clear patterns of interaction effects. Feedback
from multiple participants suggested that the information was not clearly provided (e.g. *'Interaction effects are difficult to determine [...]. I think some additions such as an arrow (up or down for conflict vs. synergy) or a texture (different hatches to denote conflict or synergy) would be very helpful for understanding interactions'* and *'There's too much information in this figure for it to be easy to understand. The asterix, while helpful to have the explanation, busies the figure'*). At the same time, participants appreciated the structure and outline using colour-coding to highlight robustness (*'I like the clear representation
of robustness tradeoffs across the three criteria'*) and completeness of information (*'I think this table shows the results of each pathways which is very informative.'*).



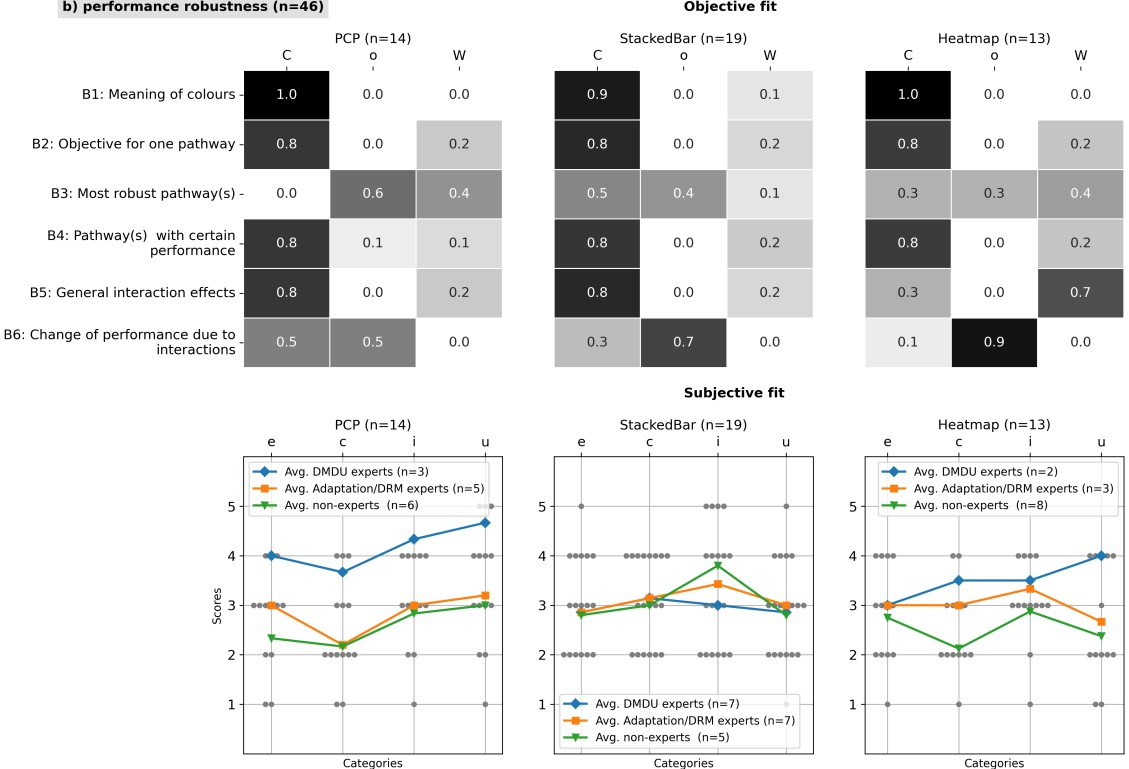

**Figure 6.** Evaluation of the dashboard for the second theme of analysis ('How do the pathways options perform?') based on the inputs from the users (n=46). Left: Evaluating the objective fit based on the share of correct answers (C) compared to wrong (W) and partially correct answers (o) (for the full set of questions refer to Table C1 in the Appendix). Right evaluating the subjective fit, differentiated in how easy they find the visualisation (e), how confident they are about the made choice (c), if the had enough information (i) and whether they would use this visualisation type for similar problems (u).

## 3.3 Evaluation of the dashboard to support pathways timing analysis

PM (Figure 2e) were used to analyse the timing, with six questions that evaluated the objective fit, as shown in Figure 7left. Interestingly, the participants only struggled with Question C2 (hit rate 40%, n=44), which asked for the maximum number 290 of measures to be implemented in a certain scenario for any pathway. Most of the participants who gave a incorrect answer indicated a higher number of measures than actually necessary, which can be related to lack of clarification on the different markers used (e.g. *'I don't know what the filled in vs. not filled circles meant'*). It appears that participants who struggled with Question C2 did not make use of interactive options to highlight the pathways from or to a specific measure, which make the pathways of interest distinguishable from the rest. Furthermore, some might not have seen that additional information on 295 demand available in a box below the plot (*'I prefer to have the button explanation in the figure, rather than use it in a legend.'*).





In general, the evaluation of subjective fit of the pathways was positive, as shown in Figure 7right. Although most of the non-expert participants would not agree that the visualisation is easy to understand and that they are confident in their choices, the participants tended to agree that the visualisation offers enough information and that they would use such a figure for similar purposes. The most relevant challenges that participants encountered with the figure included difficulty distinguishing

between overlapping pathways, especially when several converge around the same tipping points. Some participants found it difficult to differentiate the colours, making it difficult to follow specific pathways and understand the timing of certain measures. The absence of pathway numbers and the close proximity of circles made the figure harder to navigate, with some participants unsure if empty markers represented tipping points or measures. Additionally, the reliance on visual rather than textual information and the placement of the legend added to the confusion. Some participants also struggled to understand the

goals implied by questions such as 'need to be' and a few found it difficult to comprehend the y-axis.

On the positive side, participants appreciated the visualisation's ability to clearly represent the timing of measures and tipping points once they became familiar with it. The interactive elements that allowed participants to click on the pathways for more detailed information were considered a valuable feature. The figure effectively illustrated the path dependencies and the influence of interactions on timing (e.g. *'It is easy to identify synergies'*). The design also allowed for a clear comparison of

long-term versus short-term actions (*'The concept is quite intuitive and assists in seeing long-term vs short-term actions and what is available later in the period'*). In general, the participants found the PM to be a strong communication tool to represent complex scenarios.

## 3.4 Evaluation of the dashboard to support system-level pathways analysis

### 315 3.4.1 System-level performance analysis

Evaluating the objective fit revealed challenges with the navigation of the interface and the clarity of the figure, as shown in Figure 8. Some participants (HM: n=2, PCP: n=2, SBC: n=1) filled in obviously incorrect answers in combination with a clear indication that they could not read the figures because they did not use the navigation bars of the dashboard (*'don't know, too much complexity!'* or *'No data was displayed. Did I do sth wrong? My answers are not based on any analysis.'*). One participant

noted that the interface was less easy to use for this analysis question than for others, making the analysis more complicated than necessary: *'I think it was more the fiddly interface but this section was less easy to operate and understand for me.'*. Most of the participants found the visualisations rather difficult to understand along with a similar reasoning as outlined in Sect.3.2. Furthermore, labels used to indicate combinations of pathways from different sectors felt abstract and difficult to interpret quickly (*'The row label (e.g., 1,5,3,0) can acquire some effort to understand correctly'*).

Specifically, SBC were more effective and the participants agreed that they would reuse them for similar tasks, while PCP and HM were harder to interpret, as summarised in Figure 8right. An explanation may be that the option to gradually increase the number of stacked bars to be shown helps its completeness (e.g. *'was nice to be able to show multiple combinations in one figure for robustness'*).





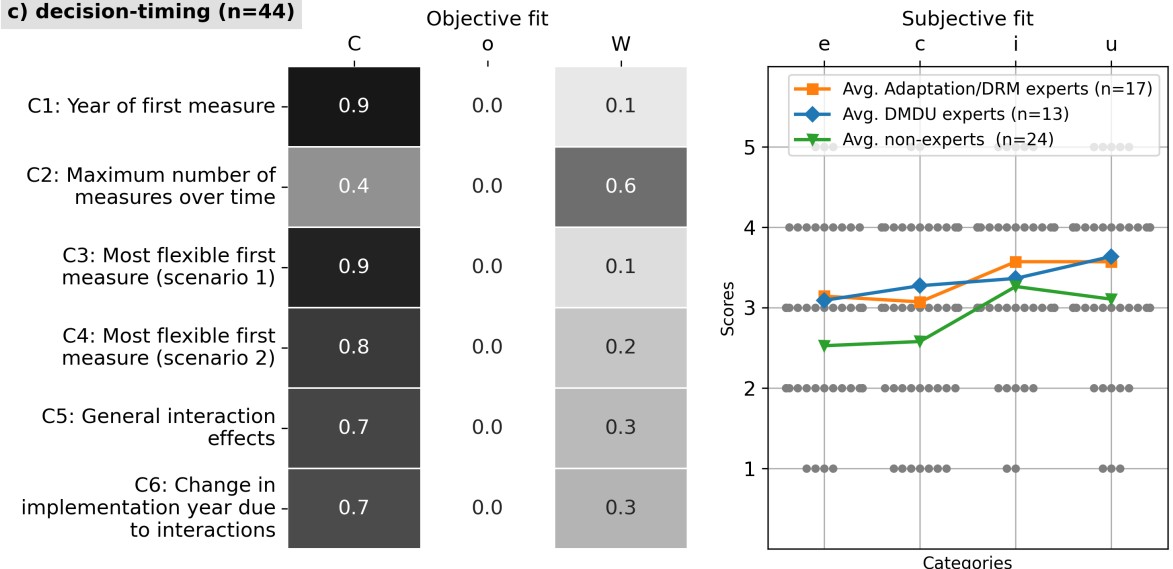

**Figure 7.** Evaluation of the dashboard for the third theme of analysis ('How do these pathways options map out in time?') based on the inputs from the users (n=44). Left: Evaluating the objective fit based on the share of correct answers (C) compared to wrong (W) and partially correct answers (o) (for the full set of questions refer to Table C1 in the Appendix). Right evaluating the subjective fit, differentiated in how easy they find the visualisation (e), how confident they are about the made choice (c), if the had enough information (i) and whether they would use this visualisation type for similar problems (u).

### 3.4.2 System-level decision-timing analysis

PM for timing analysis showed a strong objective fit, as shown in Figure 9left, although some participants reported technical problems or feeling overwhelmed by information (e.g. *'The pathway map figure is not working for me. Please disregard all answers pertaining to it (answering was mandatory).'*). Subjectively, participants valued the feature that allowed the highlighting of specific pathways, helped clarity, and made it easier to explore the integration of pathways into a broader set of combinations of pathways. The subjective fit was perceived as overall positive. The participants found several advantages in

the figure. Participants indicated that they liked the simplicity of the pathways figure, finding it less overwhelming than PM in the previous analysis theme, resulting in a similar evaluation of the subjective fit as shown in Figure 9right.

## 4 Discussion

In this study, we developed a visual analytics dashboard prototype to support pathways analysis in complex systems, with applications for multi-risk DRM and DMDU. Despite study limitations, our findings provide valuable insight into the design

process and visualisations for pathways analysis, offering lessons relevant beyond this study.



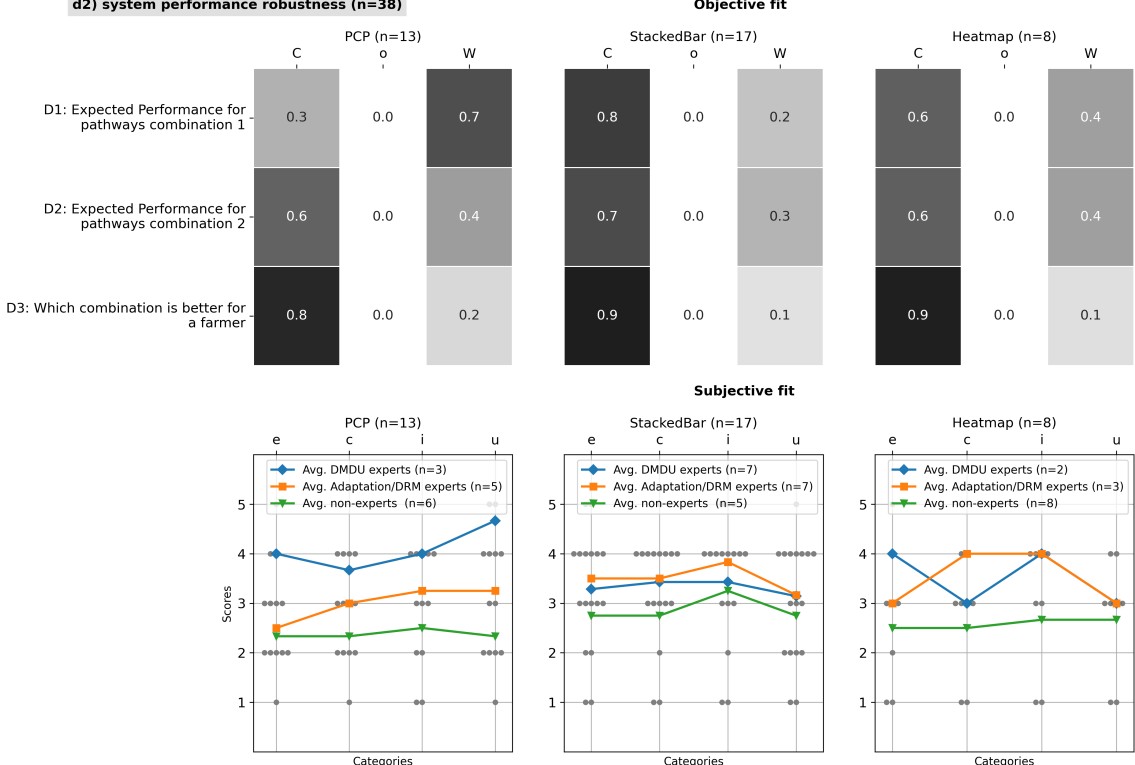

**Figure 8.** Evaluation of the dashboard for the fourth theme of analysis ('Which combinations of pathways serve multiple hazards and sectors?') based on the inputs from the users (n=38). Here, the focus is on the analysis with regards to the performance. Left: Evaluating the objective fit based on the share of correct answers (C) compared to wrong (W) and partially correct answers (o) (for the full set of questions refer to Table C1 in the Appendix). Right evaluating the subjective fit, differentiated in how easy they find the visualisation (e), how confident they are about the made choice (c), if the had enough information (i) and whether they would use this visualisation type for similar problems (u).

## 4.1 Limitations

This study has several limitations that may have impacted our findings. First, while the dashboard was designed for collaborative decision-making in a participatory modelling context, participants tested it as a standalone tool without any introductory presentation in the context of case studies. Some participants noted the need for additional context and training, indicating that such a complex topic requires more than an intuitive interface. Second, we evaluated the dashboard with 54 participants, which - while comparable to similar studies (e.g., Bautista and Carenini, 2008; Conati et al., 2014; Dimara et al., 2018; Gratzl et al., 2013; Shavazipour et al., 2021) - is still limited, especially given the varied expertise and distribution among visualisation types. However, anecdotal feedback, which is a crucial source of information on visualisation utility (Kosara et al., 2003), was consistent among participants. This suggests that the sample size may have been sufficient (Munzner, 2008). However, most





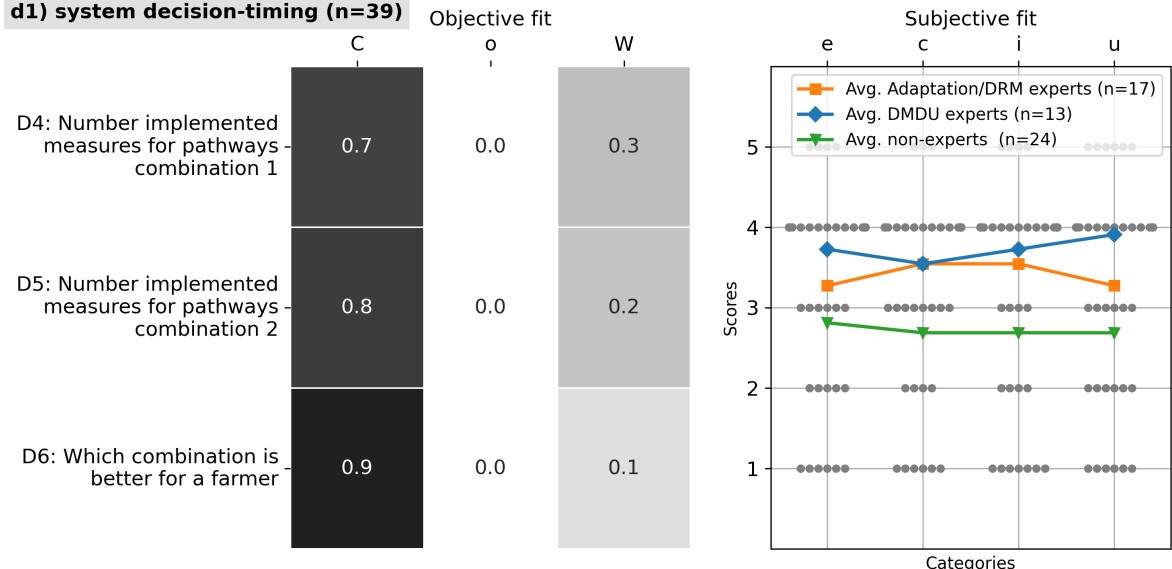

**Figure 9.** Evaluation of the dashboard for the fourth theme of analysis ('Which combinations of pathways serve multiple hazards and sectors?') based on the feedback from the users (n=39). Here, the focus is on the analysis with regards to the timing. Left: Evaluating the objective fit based on the share of correct answers (C) compared to wrong (W) and partially correct answers (o) (for the full set of questions refer to Table C1 in the Appendix). Right evaluating the subjective fit, differentiated in how easy they find the visualisation (e), how confident they are about the made choice (c), if the had enough information (i) and whether they would use this visualisation type for similar problems (u).

of the participants were researchers, while policy makers or decision makers are the primary intended users. Given that early

adopters of multi-risk DRM are often involved through research projects (Šakić Trogrlić et al., 2024), this limitation may be

acceptable. However, more tests with larger and more diverse sets of participants are needed to validate our findings. Finally,

the choice and complexity of the survey questions probably influenced the evaluation of the dashboard. We balanced simple

and complex questions following the example of Conati et al. (2014) to obtain diverse insights while keeping the survey man-

ageable, but some participants found certain questions unclear, potentially leading to confusion or errors. Evaluating decision

support tools is inherently challenging, as subjective metrics such as confidence and satisfaction can be noisy indicators of

usability (Dimara et al., 2018).

## 4.2 Design process insights

Despite these limitations, we gained meaningful insight into the design process and its results. Systematically defining visual-

isation elements, identifying users, their objectives, and their approach to finding information and matching it with available

data and visualisation types were essential during the design process. For example, the iterative refinement of analysis questions

and operations, particularly in a complex domain, confirmed the importance of continuously revisiting these design elements





(Johnson, 2004; Munzner, 2008; **?**). The survey feedback emphasised the value of involving end users throughout the design process to minimise confusion and ensure that visualisations meet their intended purpose effectively (Sedlmair et al., 2012).

Outside of the information visualisation research community, there seems to be limited application of systematic design processes. We came across multiple studies that discussed or used visualisations with potential users (e.g., Gill et al., 2020; Shavazipour et al., 2021) or mentioned fundamental design principles to adhere to (e.g., Bonham et al., 2022), but none that provided explicit reasoning for the final design or insight into the design process. Based on our positive experience, it seems vital for research communities such as multi-risk DRM or DMDU to not underestimate the value of taking the time to think

about how to use visualisations and for what purpose (Munoz et al., 2018).

### 4.3   Dashboard effectiveness

Survey results suggest that DT, PCP, SBC and PM effectively support the analysis of pathways in complex systems, while HM seems less suitable. Most of the participants answered the analysis questions accurately, demonstrating the potential of the dashboard for decision support. For some questions, e.g. question 3 in the performance robustness theme of analysis, inherent

strengths and weaknesses of different visualisation types also contributed to the quality of the responses. Ideally, users would be allowed to switch between different visualisation types for specific analysis tasks or to confirm their interpretations. For example, while PCP help explore tradeoffs across objectives, SBCs are good at comparing the overall performance across multiple objectives.

We incorporated interactive elements and a step-by-step analysis process to balance data complexity with user capacity

(Franconeri et al., 2021). Most of the participants appreciated interactive elements to allow them to explore different scenarios and analysis. The ability to hover or click to explore options in greater detail allowed users to simplify complex information. For example, in performance analysis, participants appreciated that hovering provided additional information, which could otherwise have been overwhelming if presented simultaneously. Additionally, evaluation suggests that users grew more confident with specific visualisation types (e.g. PM or SBC) across the individual themes of analysis despite added complexity.

However, feedback highlighted challenges related to information density. Multiple participants felt that dashboard visualisations showed too much information, while others found certain elements lacking sufficient information, particularly with regard to key concepts new to most survey participants (e.g., 'synergies', 'robustness') or did not fully utilise these features, suggesting the need for clearer instructions on how to use interactive elements to improve user experience and understanding. Several respondents suggested implementing storytelling techniques or scenario-based examples to make the analysis more

relatable, which suggests that the effectiveness of the chosen visualisations and the dashboard can still be improved.

### 4.4   Contributing to the fields of multi-risk DRM and DMDU

This dashboard prototype, along with the user feedback collected, provides contributions on the use of visualisations and dashboards in the emerging field of multi-risk disaster risk management and DMDU. Most of the applied visualisations, such as HM, PCP, and PM, are already widely used within the DMDU community (Hadjimichael et al., 2024). Our study provides

insights into the strengths and limitations of each visualisation type for users with varying degrees of expertise. By evaluating





these visualisations in a structured environment, we contribute evidence on the utility and potential pitfalls of each approach, supporting their adaptation in future DMDU applications. This study also emphasises the value of interactive visualisations for DMDU, such as our dashboard, providing users with options to explore details, interpret properties (e.g., tipping points within PM), and adjust the analysis focus (e.g., filtering by scenarios or time horizons). The interactive elements proved

beneficial in helping users manage the complexity of the data by enabling a customised exploration, thus enriching the decision-making process. This study joins a small but growing body of work demonstrating the benefits of interactive visualisation in DMDU, such as Bonham et al. (2024), which developed a dashboard for evaluating water management strategies under different robustness criteria.

At the same time, this study offers a starting point to discuss and improve the toolset for policy analysis in the context of

multi-risk DRM. The demand for DRM approaches that consider cross-sectoral, multi-hazard interactions over time is gaining traction (IPCC, 2022; Simpson et al., 2021; Thaler et al., 2023; UNDRR; Ward et al., 2022; Westra and Zscheischler, 2023) and there is a growing body of conceptual guidance to do so to support decision-making (e.g., de Angeli et al., 2022; Hochrainer-Stigler et al., 2023; Schlumberger et al., 2022). However, our experience developing this dashboard highlights a persistent gap (Boon et al., 2022): While decision makers are encouraged to consider interconnected risks and interacting strategies,

visualisation tools capable of clearly illustrating these complex interactions to help a decision maker to choose between two DRM options remain scarce.

This dashboard prototype and our findings from the iterative design and evaluation process could serve as a starting point for developing (better) multi-risk DRM decision support tools. Specifically, insights from our design process offer a basis for discussing and identifying (additional) key analysis questions relevant to multi-risk DRM, while the dashboard offers visual

elements suitable to answer these questions effectively. In this study, we assumed decision makers would first tackle sector-specific risk strategies before incorporating multi-sectoral interactions. This approach, which progresses from simpler to more complex analyses, proved effective and may offer a practical approach for supporting decision-making in multi-risk DRM.

## 5   Conclusions & Recommendations

This study presents a novel visual analytics dashboard prototype tailored to support pathways analysis in complex, multi-risk

decision-making contexts, specifically within Disaster Risk Management (DRM). Using a systematic approach of iterative design, we developed a dashboard that addresses key steps in the analysis of pathways in complex systems, such as exploring pathway options, evaluating the robustness of performance, and visualising decision timing. Feedback from 54 participants at various levels of expertise provided information on the utility, strengths, and limitations of the dashboard, revealing both the potential and areas for improvement in visualisation-based decision support for DRM.

The findings indicate that DT, PCP, SBC, and PM are effective for analysing pathways within complex systems. These visualisations enable users to engage with DRM data, facilitating a comparative analysis of pathway options across dimensions like performance and timing. Participants valued the dashboard's interactivity, which allowed them to investigate different scenarios, explore specific measure sequences, and access additional details on demand. However, feedback also highlighted





challenges with information overload, where participants felt overwhelmed by the volume of data or noted a lack of context

for certain elements.

This study contributes to the Decision-Making Under Deep Uncertainty community by offering empirical evidence on the effectiveness of specific visualisations in the analysis of pathways. The prototype dashboard addresses a gap in DRM decision support tools by enabling multi-criteria and multi-risk analysis through interactive, user-centred design. However, improving the objective and subjective fit of the dashboard by addressing survey feedback is an important next step. In particular, while

the dashboard effectively supports pathway comparison in terms of sequence, performance, and timing, participants noted that it offers limited insight into the underlying dynamics that explain pathway outcomes. This explanatory gap limits the utility of the dashboard as a decision support tool, particularly for users who need to understand the trade-offs and synergies behind different choices. Incorporating additional visualisations, such as time series graphs, could clarify how pathways evolve and why specific outcomes occur.

Moreover, adapting this prototype to a flexible, generalisable framework could allow it to be tailored for different datasets, criteria, and design choices, broadening its applicability. Although designed for DRM, the flexible structure of the dashboard suggests that it could be adapted for use in other domains, such as climate-resilient development, where decision makers also face complex, multi-criteria, and uncertain environments (Di Fant et al., 2025 (in review; Langendijk et al., 2024). Studying how learning and decision making evolve around such a tool would be valuable, especially as different stakeholders can bring

diverse perspectives and criteria, often requiring negotiation to identify optimal DRM pathways for the system as a whole (Gold et al., 2022; Smith et al., 2019). In general, this dashboard prototype demonstrates the potential of visual analytics to support the analysis of DRM pathways by managing the complexity of multidimensional data and by facilitating a nuanced understanding of the pathways options and their implications. With improvements in accessibility, guidance, and adaptability, the dashboard could serve as a valuable tool for decision makers navigating uncertain futures across sectors. Recognising

and managing the complexity of multiple risks and actors is becoming increasingly important in light of climate change and socioeconomic developments.

*Code and data availaility.* The code for the dashboard will be made avaialable on Github upon acceptance of this manuscript along with the collected data from the survey. They can be accessed here: https://github.com/JuliusSchlumberger/PathwaysAnalysis_Dashboard

*Author contributions.* We use CRediT to distinguish authors' contribution. **Julius Schlumberger**: Conceptualisation, Data Curation, For-

mal Analysis, Investigation, Methodology, Software, visualisation, Writing (all lead).**Robert Šakić Trogrlić**: Investigation (supporting), Methodology (supporting), Writing - Review & Editing (equal), Supervision (equal). **Jeroen Aerts**: Conceptualisation (supporting), Writing - Review & Editing (equal), Supervision (equal). **Jung-Hee Hyun**: Methodology (supporting), Writing - Review & Editing (equal), Supervision (equal). **Stefan Hochrainer-Stigler**: Methodology (supporting), Writing - Review & Editing (equal), Supervision (equal). **Marleen de Ruiter**: Conceptualisation (supporting), Writing - Review & Editing (equal), Supervision (equal). **Marjolijn Haasnoot**: Conceptualisation

(supporting), Writing - Review & Editing (equal), Supervision (equal) , visualisation (supporting).



*Competing interests.* The authors have the following competing interests: Marleen de Ruiter and Robert Šakić Trogrlić are editors of the Special Issue we are submitting this manuscript to.

*Acknowledgements.* JS, RST, JHH, SHS, MCdR, and MH received support from the MYRIAD-EU project, which received funding from the European Union's Horizon 2020 research and innovation programme under grant agreement No. 101003276. MCdR also received support from the Netherlands Organisation for Scientific Research (NWO) (VENI; grant no. VI.Veni.222.169). JA has been supported in this work by the COASTMOVE ERC Grant, Grant No. 884442. A portion of the research discussed in this report was carried out during the Young Scientists Summer Programme (YSSP) at the International Institute for Applied Systems Analysis (IIASA) in 2023. We want to thank all 21 participants in our group discussions and semi-structured interviews along with the 54 participants of the survey to test the dashboard whose contribution was critical for meaningful research. Finally, Julius Schlumberger acknowledges the contributions by Dana Stuparu and Sarah Wright, who volunteered to discuss early versions of the visualisations and provided valuable feedback and ideas.



## Appendix A: Terminology for analysis operations

**Table A1.** Terms for analysis operations and their definition as suggested in Brehmer and Munzner (2013) used in this study.

| | Term | Definition | Source |
|---|---|---|---|
| **How?** | Arrange | Arrange refers to the process of organising visualisation elements spatially | Brehmer and Munzner (2013) |
| | Change | Change pertains to alterations in visual encoding. | Brehmer and Munzner (2013) |
| | Filter | Given some concrete conditions on attribute values, find data cases satisfying those conditions. | Amar et al. (2005) |
| | Overlay | Superimpose one entity on top of another so as to affect a composite appearance while still retaining the separability of each component layer. | Mullins and Treu (1993) |
| | Select | Determine a set of objects to be manipulated, enabling highlighting, annotation, filtering, or details-on-demand. | Heer and Shneiderman (2012) |
| **Why?** | Browse | Explore the system with no specific purpose other than discovering what is available. The user is inserted into various different contexts. | Mullins and Treu (1993) |
| | Compare | Examine the characteristics or qualities of two or more objects or concepts for the purpose of discovering similarities or differences. | Mullins and Treu (1993) |
| | Explore | Explore entails searching for characteristics without regard to their location, often beginning at an overview level of the visualisation [37]. | Brehmer and Munzner (2013) |
| | Identify | Recognise the nature of an object or indication according to implicit or predetermined characteristics | Mullins and Treu (1993) |
| | Lookup | Given an object, determine a specific property of that object. | Casner (1991) |

## Appendix B: The final set of information visualisations

When developing the visualisations, we took into account multiple guiding principles. Where possible, our goal was to use two different coding channels to convene the key information. As such, we used colours and different patterns to distinguish between different measures, or colours in combination with annotations or information on demand to obtain information about the performance robustness of pathways. Also, we use descriptive figure titles to allow users to easily deduce which (sub)-dataset is currently visualised. For the choice of the colour scheme, we took into consideration the potential use context of the dashboard: multiple stakeholders would analyse their specific pathways options before coming together to investigate synergies and trade-offs across sectors and risks. We identified objectives as the core element of the analysis that should be recognisable across the different steps. As such, we chose the colour schemes per sector in a way that they can be combined across the sectors without leading to confusion by changed colour-schemes.





## B1    Visualisations for the Pathways Options

To address the first set of questions regarding the pathways options, we identified an interactive Decision Tree (DT) as the most suitable visualisation to explore specific measures, their characteristics and their relevance as short-term actions or long-term

options. The focus of the visualisation is on learning about the (different) sequences that we consider as pathways options, which is consistent with the typical purpose of DT to represent hierarchical structures in data sets (Shneiderman, 1996). In addition, information on the characteristics of the measure can be obtained on demand. In line with recommended practice, we distinguish candidates of interest using two coding channels: colour and pattern (in that case, button) as shown in Figure B1 (Munzner, 2014).

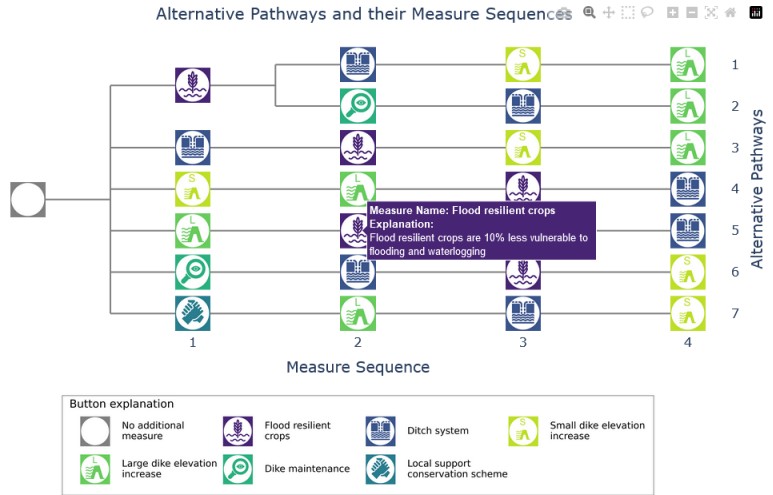

**Figure B1.** visualisation of pathways options by means of a tree map. Interactive components allow to get information on demand.

## B2    Visualisations for the Pathways Performance on sectoral and system level

We identified a set of alternatives that meet the need to provide information on the performance of the pathway options. Stacked Bar Charts, Parallel Coordinate Plots, Heatmaps. In the following, we will present each visualisation type and show screenshots.

**Parallel Coordinates Plots** (PCP) can be used to encode correlations between multiple keys. It has been recognised as a

useful tool for high-dimensional data (Itoh et al., 2017; Siirtola, 042000). Correlations across ordered attributes of multiple keys are shown by polylines that intersect multiple parallel y-axis. PCPs are useful if the number of items (options) and keys (objectives) is not too large, avoiding occlusion and complexity of analysis (Dzemyda et al., 2013; Munzner, 2014). At the same time, interactive PCPs allow users to query the information. A very common manipulation approach is filtering based on a range of interest for one (or multiple) y-axis (Siirtola, 042000). Additionally, colour hue can be used to encode a specific

key (e.g., options) to make the lines more distinguishable. We also used colour to distinguish between the performance of the



pathway options with and without interactions (Figure B2). PCPs are also capable of scaling the number of keys by increasing the number of parallel coordinates relevant for system-level analysis.

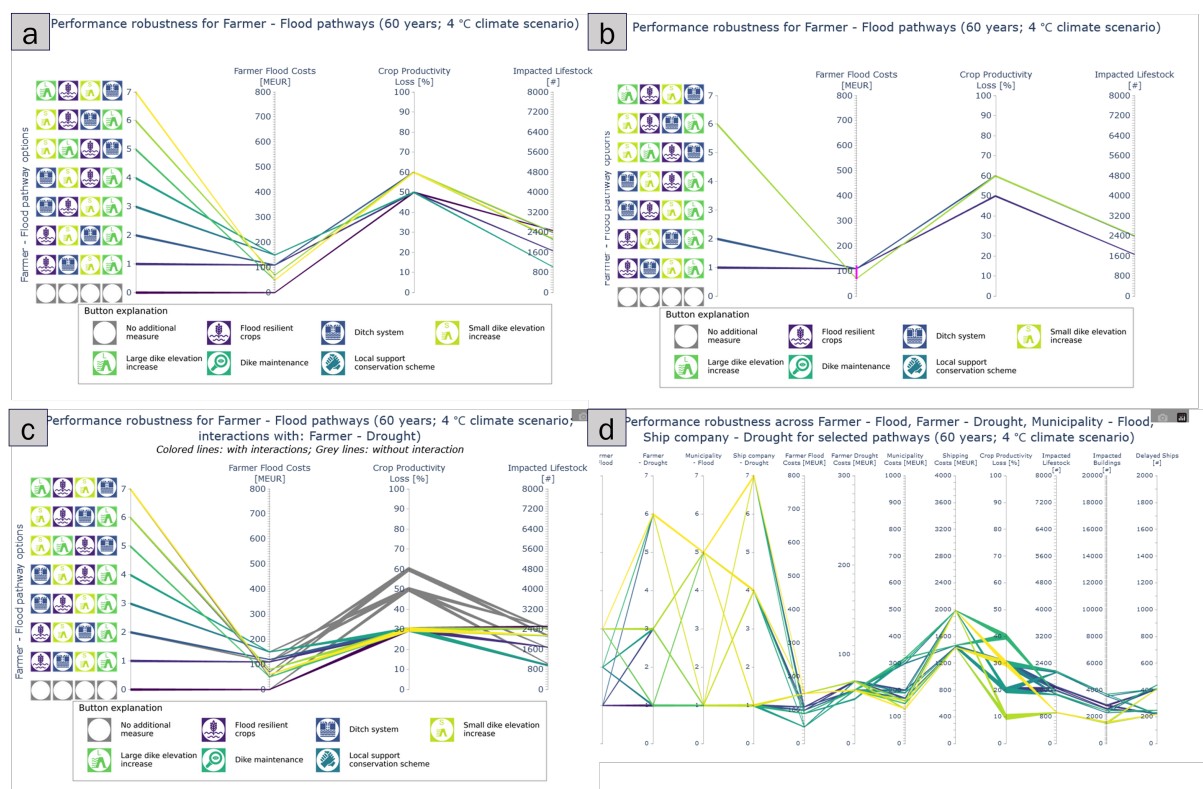

**Figure B2.** visualisation of pathways performance robustness using Parallel Coordinate Plots. Selecting acceptable ranges across multiple objective axes allows to filter pathways options (b). Changes in performance due to interactions can be distinguished by gray polylines and an annotation explanation the interpretation (c). Analysis across sectors and risks can be done by extending the set of parallel axes and pathways (combinations) (d).

**Stacked Bar Charts (SBC)** can be used to encode the sums of key attributes while also providing insight into how these attributes contribute to the total (Gratzl et al., 2013; Streit and Gehlenborg, 2014). By combining multiple stacked bars in parallel, the absolute differences for the values with respect to one key can be encoded in terms of the length of the stacked bar, while the relative importance of the values with respect to the other key can be encoded by the colour hue (Hindalong et al., 2020). SBC can be useful when the number of key attributes to stack is limited to facilitate distinction (Indratmo et al., 2018; Munzner, 2014). We implemented SBC as shown in Figure B3. We used dimensional stacking to keep the information on different objectives distinguished by different. The reason being that stakeholders might be interested in the overall performance or the performance with regards to specific. Interaction effects are indicated by pattern. Additional information is available on demand, offering insight into the actual performance, comparison of the interaction effects, and combination of pathways





options. At the system level with many more combinations of options to compare, we decided to order the stacked bars with respect to the total length, to support the analysis of the task (Gratzl et al., 2013).

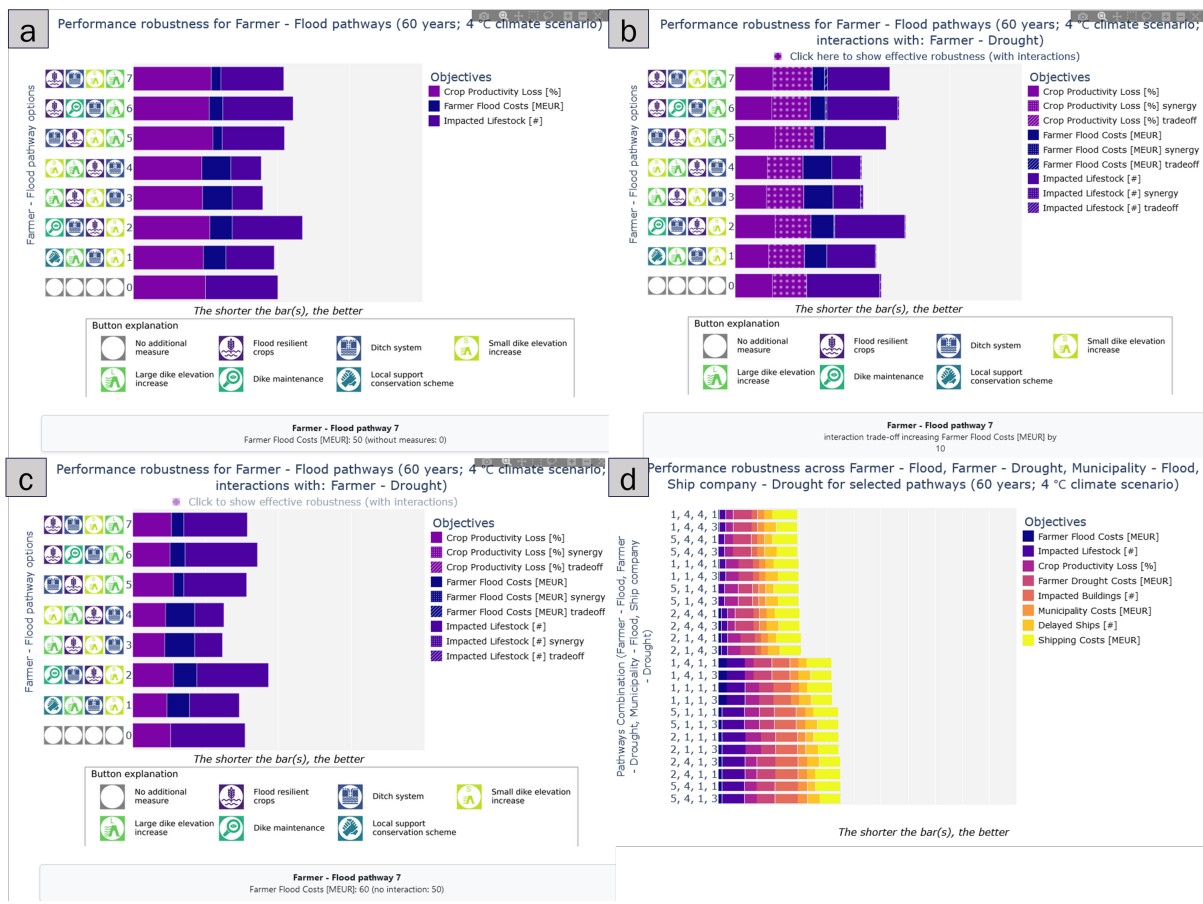

**Figure B3.** visualisation of pathways performance robustness using Stacked Bar Charts. Bars are colour-coded for different objectives and the performance is coded by means of the length of the (stacked) bars. Hovering over different bars offers information on demand (a). Interactions are shown by means of additional bar elements, where trade-off and synergy effects are coded by means of different patterns (b). These additional elements can be toggled off in case users want to compare the effective performance across pathways options (c). Analysis across sectors and risks can be done by extending the set of pathways (combinations) along the y-axis. To facilitate the analysis, the pathways combinations are sorted by means of total length of the stacked bars (d).

**Heatmaps** (HM) can be used to encode quantitative value attributes for two categorical keys by arranging colour-coded values in a matrix form (Munzner, 2014). Defining the colour hue based on the normalised values per key attribute allows the use of a common colour scheme across attributes of one of the keys (Shavazipour et al., 2021). HM can have high information density while still providing effective high-level summaries (Munzner, 2014). They can be used to identify trends between attributes of one key or general clusters (Hindalong et al., 2020). Instead of interactive elements, we used annotations to





provide additional information relevant for the analysis. As such, colour coding gives some insight into general patterns, and
annotations offer insight into the effective performance and effects of interactions (if applicable). HMs similar to PCP are
relatively well scalable allowing for additional objectives keys and option combinations. We implemented HM as shown in
Figure B4.

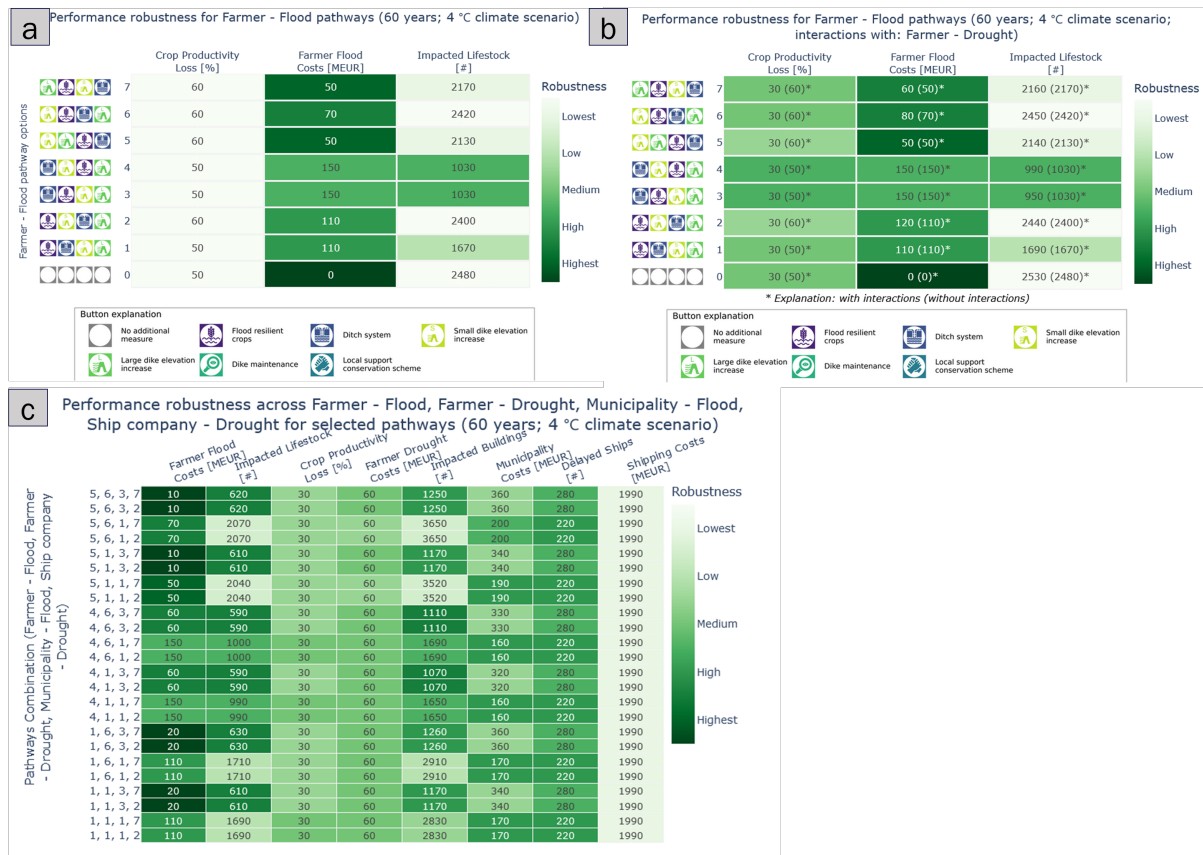

**Figure B4.** visualisation of pathways performance robustness using Heatmaps. Performance robustness is represented by the colour-coding
and the annotation for each pathways option (a). Effects of interactions are shown by comparing the robustness performance with and without
interactions (b). Analysis across sectors and risks can be done by extending the set of pathways (combinations) along the y-axis and number
of objectives along the x-axis (d).

## B3   Visualisations for the Pathways Timing

To address the questions of interest about the timing of adaptation tipping points, we identified Pathways Maps (PM) as the
promising option. PM can be used to visualise alternative sequences of decisions or actions over time, often in the context of
adaptive management or long-term planning under uncertainty (Haasnoot et al., 2012, 2024). This figure represents a 'Metro-
map' through time (starting at the left, moving to the right). The points where lines split or intersect indicate key moments



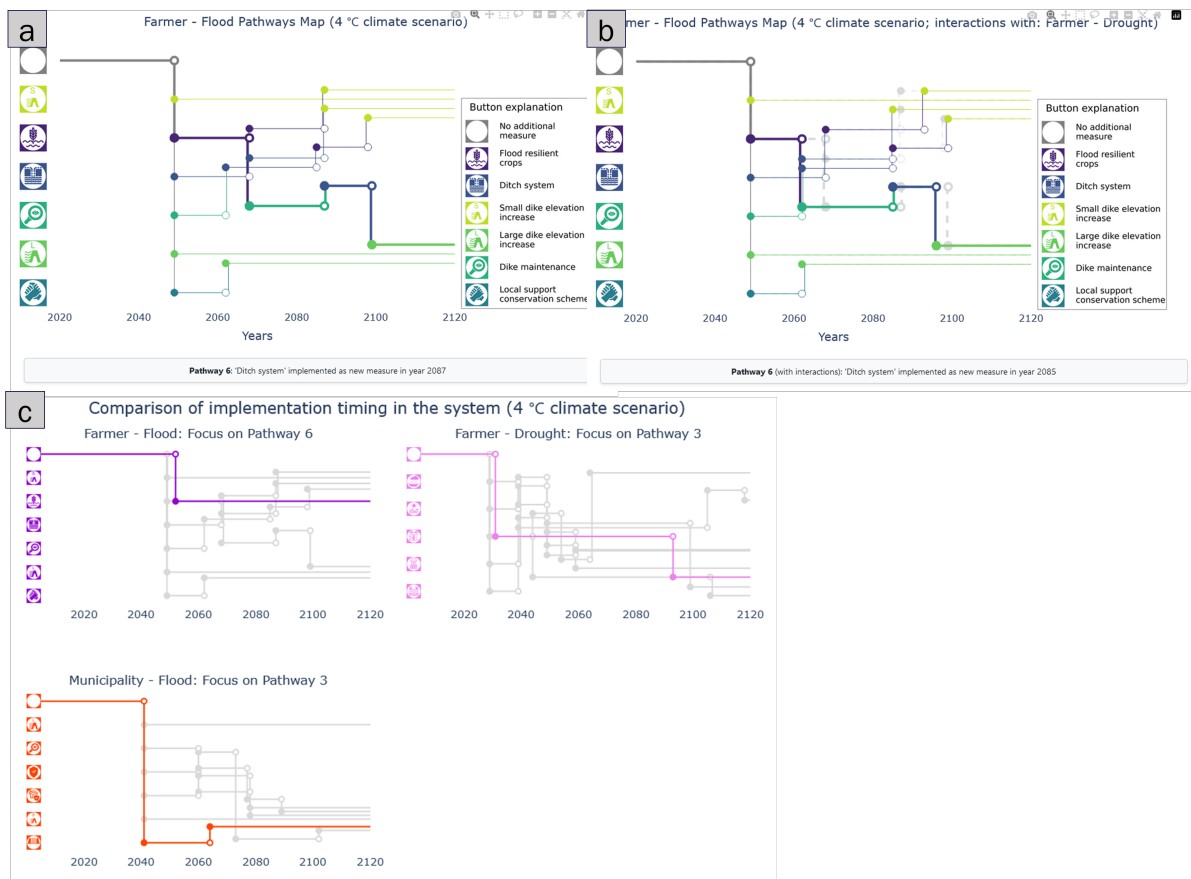

**Figure B5.** visualisation of pathways decision-timing using Pathways Maps. Following a specific strategy, the timings of new measures are shown. The interactive character of the figure allows users to select a marker in the plot to see what future options are from this point onwards or what sequences have led to the given decision-point in time (a). Effects of interactions are shown by overlaying pathways maps with and without interactions (b). Analysis across sectors and risks can be done for one pathways combination at a time. Comparing the timings of the selected pathways compared to a pathways maps without interactions gives insight into trade-offs and synergies regarding the timing (d).

where a decision is needed to either stay on the current path or switch to a new one. This ensures that the chosen pathway remains effective as circumstances evolve. Each branch represents a potential future trajectory based on choices made at times

when the system requires additional measures being implemented (adaptation tipping points). PM are particularly useful when stakeholders need to understand both short-term and long-term options, as well as how decisions made now can influence future flexibility. We implemented PM as interactive visualisations, allowing one to highlight a specific pathway of interest or a short-term action to explore how possible future options plan out (Figure B5. Furthermore, the visualisation offers information on demand regarding the specific adaptation tipping points to support the analysis. By overlaying PM for cases with and without

interactions, effects on the timing of adaptation tipping points are detectable. To account for specific combinations of pathways



for the system-level analysis, instead of integrating pathways into a system-level PM as done by (Schlumberger et al., 2022), we develop separate maps for separate actors to manage complexity of the analysis.

## Appendix C:  Evaluating the objective and subjective fit

### C1   Survey questions

Table C1: Overview of all questions, correct answers or inputs of the survey

| General Questions before start of the survey | | |
| --- | --- | --- |
| G1 | Do you have any visual impairments or conditions that might influence the way you perceive visual content? | Options: [Yes: specify / No / I don't know / I don't want to share] |
| G2 | What is your field of work? | Options: [Research / Public Administration / Private Sectore / Other] |
| G3 | What are your areas of expertise (use key terms and separate by ';') | Free-text |
| G4 | How often do you use visualisations for analysis? | Options: 1-5 Likert scale (never – every day) |
| G5 | What is your experience with the following visualisation techniques? ['SBC', 'PCP', 'H', 'Pathways Map'] | Options for each viz type: 1-5 Likert scale (never – every day) |
| | What are the pathways options? | |
| A1 | How many pathway alternatives do you have? | 7 |
| A2 | How many alternative pathways start with measure 'flood resilient crops'? | 2 |
| A3 | Which measure is considered most often as the long-term measure (being implemented at a later stage)? | large_dikes |
| A4 | Which first implemented measure offers the most flexibility with regards to future options? | Flood Resilient Crops |
| A5 | I find this figure easy to understand | Options: 1-5 Likert scale (totally disagree – totally agree) |
| A6 | I am confident that I read this figure correctly to inform my answer-choice | Options: 1-5 Likert scale |
| A7 | This visualisation provides enough information to justify your answer | Options: 1-5 Likert scale |
| A8 | I would use this visualisation for similar problems | Options: 1-5 Likert scale |
| A9 | Please briefly describe one or two challenges you had when reading the figure (if any) | Free-text |





| | | |
|---|---|---|
| A10 | Please briefly describe one or two things you find useful about this figure (if any) | Free-text |
| **How do the pathways options perform?** | | |
| B1 | What do the colours represent in the figure? | Depends on viz type: pathway, robustness, objectives |
| B2 | How much Crop Productivity Loss [%] do we expect for Pathway 5 over a time horizon of 60 years in the 4 °C climate scenario with no pathway interactions considered? | 60 |
| B3 | In the 4 °C climate change scenario, which pathway(s) is most robust at the time horizon of 60 years with no pathway interactions considered? | [3,4] |
| B4 | Which pathway(s) results in the highest Impacted Lifestock after 100 years in a 1.5 °C climate scenario with no pathway interactions considered? | [0] |
| B5 | When accounting for the presence of Farmer - Drought interactions, do we experience more synergy or more trade-off effects in a 1.5 °C climate scenario over the next 60 years? | Synergies |
| B6 | When accounting for the presence of Farmer - Drought strategies, which pathway(s) show the best robustness regarding Crop Productivity Loss in a 4 °C climate scenario over the next 60 years? | [0,1,2,3,4,5,6,7] |
| B7 | I find this figure easy to understand | Options: 1-5 Likert scale |
| B8 | I am confident that I read this figure correctly to inform my answer-choice | Options: 1-5 Likert scale |
| B9 | This visualisation provides enough information to justify your answer | Options: 1-5 Likert scale |
| B10 | I would use this visualisation for similar problems | Options: 1-5 Likert scale |
| B11 | Please briefly describe one or two challenges you had when reading the figure (if any) | Free-text |
| B12 | Please briefly describe one or two things you find useful about this figure (if any) | Free-text |
| **How do these pathways options map out in time?** | | |
| C1 | In which year is the first measure needed in a 1.5 °C climate scenario with no pathway interactions considered? | 2052 |
| C2 | What is the maximum number of measures that need to be implemented in one pathway in a 1.5 °C climate scenario over the 100 years with no pathway interactions considered? | 2 |
| C3 | In a 1.5 °C climate scenario, which first implemented measure offers the most flexibility with regards to future options? | Flood Resilient Crops |
| C4 | In a 4 °C climate scenario, which first implemented measure offers the most flexibility with regards to future options? | Flood Resilient Crops |



| C5 | When accounting for the presence of Farmer - Drought interactions, what is the general effect on the timing of measure implementation compared to the case without interactions in a 4 °C climate scenario? | earlier |
|---|---|---|
| C6 | When accounting for the presence of Farmer - Drought interactions, by how many years does the implementation of 'Large Dike elevation increase' in pathway 6 shift in a 4 °C climate scenario compared to the case without interactions | -3 |
| C7 | I find this figure easy to understand | Options: 1-5 Likert scale |
| C8 | I am confident that I read this figure correctly to inform my answer-choice | Options: 1-5 Likert scale |
| C9 | This visualisation provides enough information to justify your answer | Options: 1-5 Likert scale |
| C10 | I would use this visualisation for similar problems | Options: 1-5 Likert scale |
| C11 | Please briefly describe one or two challenges you had when reading the figure (if any) | Free-text |
| C12 | Please briefly describe one or two things you find useful about this figure (if any) | Free-text |

Which combinations of strategies serve multiple hazards and sectors?

| D1 | Looking at Pathways Performance with the pathway combination Farmer Flood - Pathway 1, Farmer - Drought Pathway 5, Municipality - Flood Pathway 6 and Shipping - Drought Pathway 0: what are the expected Farmer - Flood Costs in a 4 °C climate scenario? | Different options: [10 / 110 / 150] |
|---|---|---|
| D2 | Looking at Pathways Performance with the pathway combination Farmer Flood - Pathway 1, Farmer - Drought Pathway 5, Municipality - Flood Pathway 3 and Shipping - Drought Pathway 0: what are the expected Farmer - Flood Costs in a 4 °C climate scenario? | Different options: [0 / 20 / 30] |
| D3 | Which of the two considered Municipality Flood Pathways is more attractive from a Farmer - Flood perspective in a 4 °C climate scenario? | 3 |
| D4 | Looking at Pathways Maps with the pathway combination Farmer Flood - Pathway 1, Farmer - Drought Pathway 5, Municipality - Flood Pathway 6 and Shipping - Drought Pathway 0: how many measures are implemented for Farmer - Flood Pathway 1 in a 4 °C climate scenario? | 2 |
| D5 | Looking at Pathways Maps with the pathway combination Farmer Flood - Pathway 1, Farmer - Drought Pathway 5, Municipality - Flood Pathway 3 and Shipping - Drought Pathway 0: how many measures are implemented for Farmer - Flood Pathway 1 in a 4 °C climate scenario? | 1 |
| D6 | Which of the two considered Municipality Flood Pathways is more attractive from a Farmer - Flood perspective in a 4 °C climate scenario? | 3 |
| D7 | I find this figure easy to understand | Options: 1-5 Likert scale |



| D8 | I am confident that I read this figure correctly to inform my answer-choice | Options: 1-5 Likert scale |
|---|---|---|
| D9 | This visualisation provides enough information to justify your answer | Options: 1-5 Likert scale |
| D10 | I would use this visualisation for similar problems | Options: 1-5 Likert scale |
| D11 | Please briefly describe one or two challenges you had when reading the figure (if any) | Free-text |
| D12 | Please briefly describe one or two things you find useful about this figure (if any) | Free-text |

## C2  Classification of participants according to their expertise

Survey participants were asked to self-describe their expertise. We used this to investigate to what degree prior experience with the concepts is affecting the objective and subjective fit of the visualisations. The expertise attributes that were used to assign participants to specific expert groups are summarised in Table C2.

**Table C2.** Overview of expert groups and key expertise attributes that are distinctive for allocating participants

| Expert group | Distinctive expertise attributes |
|---|---|
| DMDU | Decision making under deep uncertainty, scenarios, pathways |
| Climate Change Adaptation, DRM | Climate adaptation, multi-hazards, flood adaptation, statistical modelling, DRM, risk management |
| Other | General topics without focus on uncertainty or climate adaptation, includes fields like economics, water quality, food systems |



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
