# Peer review of "A Pathways Analysis Dashboard prototype for multi-risk systems"

_EGUsphere, 2024_

## Author Response (AR1)

**Reviewer 1**

This paper introduces a prototype dashboard designed to support pathways analysis in multi-risk Disaster Risk Management (DRM). Prior to selecting visualization types, a literature review was conducted, with particular attention given to terminology. Additional details are provided in Annex A. The paper thoroughly explains the design process, methodology, and approach to selecting visual representations. Figure 1 and Table 1 help illustrate this approach, while Annex B provides further information on the chart type trials conducted in this study. The prototype dashboard was evaluated through a survey completed by over 50 participants. The survey questions are available in Annex C. Section 3 presents the survey results and assesses the performance of each visualization type. Lessons learned, key insights, and study limitations are discussed in Section 4, "Discussion."

We would like to thank the Reviewers for their interest in our work and for carefully reading our manuscript; we greatly appreciate their insightful comments, which contribute to increasing the manuscript's robustness and improving its quality. In the following, we provide a point-by-point reply to the general and specific comments raised. Changes to the text are highlighted.

**Overall Comments**

line 404 states that "this study offers a starting point to discuss and improve the toolset for policy analysis in the context of multi-risk DRM," while line 432 transitions directly to the conclusion: "The prototype dashboard addresses a gap in DRM decision support tools by enabling multi-criteria and multi-risk analysis through interactive, user-centred design." This reviewer believes that this point could be further expanded or discussed in greater depth.

We thank the reviewer for spotting this error. In the revised version, we clarify in the conclusion that the dashboard serves as a starting point by adjusting the sentences starting in line 432 as follows:

"This study contributes to the Decision-Making Under Deep Uncertainty community by offering empirical evidence on the effectiveness of specific visualisations in analyzing pathways. The prototype dashboard presents a first attempt at addressing the gap in DRM decision support tools regarding multi-criteria and multi-risk analysis through interactive, user-centred design. However, improving the objective and subjective fit of the dashboard by addressing survey feedback is an important next step." (change R1-M1)

When printed, some figures—particularly those containing multiple sub-figures or screenshots—have text that appears too small and difficult to read. A revision of these figures is recommended.

We thank the reviewer for this valuable feedback. We've revised the figures by increasing the font size of labels, legends, and annotations. (change R1-M2)

The URL of the dashboard is provided as www.pathways-analysis-dashboard.net (line 130). However, the URL leads to the survey version of the dashboard rather than a freely explorable version.

It would also be helpful to introduce the URL earlier in the paper or highlight it more clearly.

We thank the reviewer for this good suggestion. We updated the dashboard so that it still includes the survey environment it was used in, but now allows viewers to navigate freely between the pages without the requirement to fill in the survey. We've added the link at multiple locations.

- In line 3: "This study introduces a visual analytics dashboard prototype
   (<a href="https://www.pathways-analysis-dashboard.net/">https://www.pathways-analysis-dashboard.net/</a>) designed to support pathways analysis for multi-risk Disaster Risk Management (DRM)." (change R1-M3)
- In Line 50: "In this study, our aim was to design and evaluate a visual analytics dashboard (<a href="https://www.pathways-analysis-dashboard.net/">https://www.pathways-analysis-dashboard.net/</a>) tailored for analysing pathways in multirisk settings." (change R1-M3)
- In line 126: "When developing the interactive dashboard (<a href="https://www.pathways-analysis-dashboard.net/">https://www.pathways-analysis-dashboard.net/</a>) and integrating fit-for-purpose visualisations, we focused on two components:" (change R1-M3)

**Minor comments**

**Line 96**: "The bold terms in Table 1 used for the description of the analysis operations are based on Brehmer and Munzner (2013) (definitions in Table A)." However, apart from the first column/row, there do not appear to be any bold terms in Table 1. Do you mean italicized terms instead?

Thanks for spotting this. In the text, we should have referred to the italicized terms indeed. We corrected the text accordingly: "The italicized terms in Table \ref{tab01} used for the description of the analysis operations are based on \citet{Brehmer.2013} (definitions in Table \ref{A:step2})." (change R1-m1)

**Line 129**: Consider adding footnotes to provide context on Dash and Plotly for readers unfamiliar with these tools.

We added footnotes for Dash and Plotly similar to the Pathways Generator:

- Plotly: <a href="https://plotly.com/">https://plotly.com/</a>
- Dash: https://dash.plotly.com/ (change R1-m2)

**Line 244**: The word "Question" is capitalized for B6 but appears lowercase for B3. Consider maintaining consistency.

We decided to use the lower-case version consistently and corrected it throughout the manuscript. Overall, we checked spelling and grammar for consistency. (change R1-m3)

**Line 363**: One of the references appears to be missing.

We corrected this error. No additional reference was supposed to be added; it was a small error in the latex document.

**Line 443**: There is an extra bracket in the middle of a reference.

We updated the reference. During submission, the reference was still under review. We have corrected it. (change R1-m4)

**Reviewer 2**

Thank you for inviting me to review the paper 'A Pathways Analysis Dashboard prototype for multirisk systems'. I have read the manuscript with great interest. The object of this study was to introduce a new dashboard that the authors state has been designed to support decision makers involved in multi-risk disaster risk management. The dashboard uses a pathway analysis approach coupled with a range of visualisations that can help users to navigate multi-risk DRM.

Overall, this is a well written paper, it reads fluently, and I have few minor suggestions around language and small corrections which I outline below in my minor comments. Please note that I have not exhaustively been through the references, so the authors may wish to do this. I did pick up on a few small issues with these.

There are however, in my opinion, some major revisions that will need to be addressed before the manuscript is suitable for publication in your journal. This work is missing some key detail with respect to who the target users were, how they were identified and how they were included in the evaluation of the final products. I hope that this can be rectified with some re-structuring and addition of information. I have outlined these suggestions in the 'Major Comments' section of my review.

I have detailed my comments below, I really hope that the authors find theses helpful in editing the manuscript. Please note that I am not able to edit the pdf and so have outlined my line-by-line comment below.

We are glad to hear that this reviewer found the manuscript an interesting read and we are grateful for their extensive and very helpful feedback. For some of the major comments, we split the original comment to be able to provide clear and direct responses. In the following, we provide a point-by-point reply to the general and specific comments raised. Changes to the text are highlighted.

**Major Comments:**

Methods – In general, I find that the methods section is incomplete in terms of detail. For example: In section 2.1, line 74, you mention target users. Who are these? How were they determined? Did you develop any kind of user stories to help you design the platform? For your six semi structured interviews, who were the candidates, what sectors did they come from, how were they selected? What was their level of comfort with the data and approaches? Similarly, how did you select the candidates for your focus groups (line 75), what was their make up?

We acknowledge that the information in the main text is incomplete - mainly for conciseness- as the paper captures many different aspects. We provide extensive information about the first analysis step as part of the supplemental material. Still, adding more substance regarding the process and the people involved makes sense. As such, we've rewritten lines 73 to 77 as follows:

"In the first step of the design of the pathways analysis dashboard, we defined the pathways analysis context, including identified target users and their capacities, and formulated key questions of interest. Similarly to Ruppert et al. (2013), we developed a set of user types. Based on our expertise and ongoing transdisciplinary research on multi-risk DRM, we first identified stakeholders generally involved in pathways development or risk assessment processes to aggregate specific generic characteristics of the stakeholders (e.g., capacities, questions of interest) into different user types. We calibrated and refined these user types through expert inputs from six semi-structured interviews and two 60-

minute focus groups. We engaged 21 researchers from the professional network of coauthors working on disaster risk management, risk communication, climate change adaptation, and pathway thinking in different sectors, summarised in Table \ref{A:tab01}. The interviews and workshops followed the guidelines of (Hove et al. 2005). An extensive description of this first step can be found in the Supplementary Material, including the final conceptualization of different user types." (R2-M1)

You state that potential users were based on previous studies – but I think you need more detail about who these are in this paper. As a reader coming to this work for the first time you would have no way of knowing who you were targeting.

We realize that this reference to previous studies in line 76 is confusing. They only become relevant at a later stage in step 3. As outlined in our response to the previous major comment (R2-M1), we identified abstract user types based on our experience from ongoing multi-risk research projects. This was not limited to past studies but informed by much broader knowledge. As shown in R2-M1, we've stricken these sentences from the paragraph.

Why did you select the Waal River as your case study? Is this because you were already working in this region, or because there were specific sectors there that you wanted to explore? You have included details of the case study in the supplementary material, but I would suggest that it would be beneficial to have at least a precis of some of this information in the main text.

We recognized the challenge of data analysis in a previous study, which was on a stylized version of the Waal. The challenge with developing multi-risk DRM pathways is the availability of data. Based on the identified limitation in existing tools (which we set out in this manuscript), we developed a tool-set that can support the analysis in data-rich model-driven contexts. As such, the data in this paper only serve to present a realistic sample data-set that can be used to explore the utility of interactive visualisations in answering the key analysis questions. We thus decided to use data from the previous study for illustration purposes - we are more broadly interested in the value of the visualisations and how capable they are to handle large multi-dimensional data sets. As mentioned in the previous comments, we have stricken the mentioning of the case study in the early steps of the design process and start referring to it only when it becomes relevant, namely in step 3, when it comes to required data structures relevant to capture the key information of interest.

In section 2.4.1, you have included information about the design process about the final visualisations in Appendix B. But again, I feel like there is some important information here that should be in the main body of the document, not buried in the Appendix. I would consider whether it was possible to move the entire context of this appendix into the main document.

We agree with the reviewer that we have added relevant information on the different visualisations in the Appendix. The intention was to keep the paper as concise as possible while keeping all key information close at hand (which is why we put it in the Appendix and not the Supplemental Material). However, as the reviewer raises this point and sees the benefit of adding this information to the main text, we decided to replace lines I.138 to I.157 with a concise version of Appendix B (I.473 to 537). We removed Appendix B as a result. (R2-M2)

Evaluation of the dashboard – Again, I find that there is some explanatory information missing here. In section 3.1 – why are you assessing using these 4 questions?

We noticed that this could be made clearer, so we propose the following change in I. 170:

"The set of survey questions was developed based on the general analysis questions presented in Tab. \ref{tab01}. The questions were tailored to objectively evaluate the answers for the specific case study data used. The full list of questions can be found in the Appendix \ref{A:step5}." (R2-M3)

What do you mean that most of the questions were accurately answered?

We corrected this statement: "The objective fit was assessed using four questions, see Figure 5 $\underline{a}$ , with the participants accurately answering A1 and A4." (R2-M4)

If you are only going to refer to 4 questions then I would suggest stating them in the text, rather than asking the reader to flip between the Appendix and the text for this key information. I think that this would make this section

We thank the reviewer for this comment. Instead of using the long questions in the text, which would make the long analysis even longer, we offered short versions of each question in the figures we refer to for the analysis.

You say that questions were 'less well answered for different reasons' – what reasons, how do you know what the reasons are?

We agree with the reviewer that we cannot infer causality here, but by offering evidence from the user feedback, we try to offer plausible explanations. We correct the sentences as follows: "Questions A2 and A3 were less well answered (hit rates: 60%, n = 54) for different possible reasons." (R2-M5)

With respect to the 54 people who evaluated the dashboard. How were they selected? What sectors did they represent? On Page 13, Line 254 – the respondent states that they are 'trying to put myself in the shoes of a farmer' – were any of your respondents actually farmers, given the relevance of that sector to your case study?

Thank you for spotting this. We clarify how we ended up with 54 respondents in line 170: "The coauthors shared the link to participate in the survey within their professional and personal contexts, which includes the research institutes and multi-risk projects, as well as networks from past conferences. The survey was open for 6 weeks from September to October 2024. The survey involved a broad and diverse range of 54 potential users and experts in the fields of information visualisation, disaster risk management, and pathways thinking and beyond." (R2-M6)

Discussion – In the limitations section I find the following statement surprising: 'First, while the dashboard was designed for collaborative decision-making in a participatory modelling context, participants tested it as a standalone tool without any introductory presentation in the context of case studies. Some participants noted the need for additional context and training, indicating that such a complex topic requires more than an intuitive interface.' Why didn't the authors test the dashboard in the context in which it would most likely be used, that is with the accompanying contextual information? I am concerned that without this context the respondents' responses are likely to be less useful – perhaps the authors could give their thoughts on this?

We see the point the reviewer is making. We have discussed the concern that the context of respondents' responses is less valuable, given that we did not test it in the setting it is designed for. We generally agree that this is true for the collaborative part of the designed dashboard. We have not tested how useful the tool is in settings with multiple users with different interests. On the other hand, the users that tested the dashboard for a particular purpose were relatively well capable of using the visualisations and the dashboard for its purpose of analysis. Our hypothesis is that the

more information we offer to participants, the easier it will be for them to use the visualisations and answer the questions. In our setup, we tested the dashboard in an extreme case where users had to deduce everything from the visualisations themselves. As such, we tested how the dashboard and visualisations, without any further guidance or influence, serve their purpose. We are thus convinced that the chosen approach could have been risky (if participants would have not been able to properly use the dashboard). However, in return, our approach resulted in even more substantial evidence that the dashboard could be a promising starting point to explore visual analytics for multirisk DRM analysis further. As a result of this reflection, we added further reflections in the discussion of limitations in line 342ff:

"This study has several limitations that may have impacted our findings. First, while the dashboard was designed for collaborative decision-making in a participatory modelling context, participants tested it as a standalone tool without any introductory presentation in the context of case studies. Some participants noted the need for additional context and training, indicating that such a complex topic requires more than an intuitive interface. We acknowledge that testing the dashboard without its intended contextual framing may limit the generalisability of participants' responses - particularly concerning its participatory development process and collaborative use, which remain open research questions. However, as the available multi-risk DRM pathways case study did not offer any involved stakeholders, we intentionally chose this minimal setup to test the dashboard's standalone interpretability as a form of stress-testing. The fact that many users could still use the tool effectively suggests a robustness in the design and a promising foundation for future, more contextualized applications. Second, we evaluated the dashboard with 54 participants, which - while comparable to similar studies \citep[e.g.,][]{ Bautista.2008, Conati.2014, Dimara.2018, Gratzl.2013, Shavazipour.2021} - is still limited, especially given the varied expertise and distribution among visualisation types. " (R2-M7)

My larger concern here is, however, that in this section it seems to be revealed that all participants were researchers and not either policy makers or decision makers (Page 19, Line 350). They go on to say that 'Given that early adopters of multi-risk DRM are often involved through research projects, this limitation may be acceptable'. This appears to be an incredibly research centric view of the world and I don't think that this is indeed what Šakic Trogrlic et al. said in their paper. The way that this reads to me at the moment is that its acceptable that you haven't included decision makers in the review of a tool to support decision making, because in fact researchers will be available to guide them with respect to multi-risk DRM. I suspect that this is not what you mean, I think this statement would benefit from: some clarity on what you mean by 'early adopters', how you see your visualisations being co-developed by policy partners and how this could feed into the implementation of multi-risk DRM, which we most definitely should not assume would be operationalised in partnership with research scientists.

We thank the reviewer for this comment. We clarified our intention by rephrasing I.349 to 352:

"This suggests that the sample size may have been sufficient \citep{Munzner.2008}.

However, most participants were researchers, while policy- or decision-makers are the primary intended users. This choice was deliberate, given that multi-risk decision-making remains a relatively new and complex topic \citep{SakicTrogrlic.2024} and is still largely situated in exploratory research and pilot initiatives. In line with the theory of diffusion of innovation \citep{Rogers.2003} we here report on a first test with a small and engaged audience that we reached through our networks within and beyond multi-risk DRM research. Future phases of development will require co-development with policy partners in projects that are relevant to them to ensure the tool fits operational realities and supports real-world

implementation. Finally, the choice and complexity of the survey questions probably influenced the evaluation of the dashboard." (R2-M8)

Another concern here is why the testing of this visualisation tool was not designed to target the projected end users? On Page 19 you state that 'The survey feedback emphasised the value of involving end users throughout the design process to minimise confusion and ensure that visualisations meet their intended purpose effectively'. But this contradicts the fact that you didn't seem to involve the end users that you have targeted the product to. If you have indeed included them then this is unclear to me. I very much agree with this statement: 'it seems vital for research communities such as multi-risk DRM or DMDU to not underestimate the value of taking the time to think about how to use visualisations and for what purpose' but from your manuscript I don't think that you have always clearly evidenced this approach.

Thank you for the comment. During the design process, we collected feedback on the developed visualisations from a small group of colleagues and co-authors (this information is now included in the main text as a result of our response to R2-M2). Based on those discussions, we revised the visualisations. A larger group ultimately tested the visualizations, resulting in some confusion. Our case did not lend itself to an actual participatory development process as the underlying data were drawn from a stylized case without actual stakeholders involved. We have clarified this limitation in our response to R2-M10. We further clarified this in the discussion by revising I. 363:

"In this study, we only engaged in limited feedback throughout the design process, which still offered valuable insights regarding limitations and useful elements of the visualisations. Involving actual decision-makers can even further improve the utility of the visualisations to minimise confusion and ensure that visualisations meet their intended purpose effectively \citep{Sedlmair.2012}." (R2-M9)

I think that these 'limitations' call into question the overall usefulness of this visualization tool and undermine your conclusions that what you have produced is a tool that will be beneficial in the wider context of multi-risk DRM. I think that the authors need to therefore re-visit the manuscript in the context of these questions. Perhaps this is a case of clarifying the narrative and the language to more accurately frame the work that has been done? Or perhaps there are some gaps here that need to be addressed. I'm not entirely sure from that what I've read which of these is the case.

Minor Comments:

Abstract -

Page 1, Line 1: 'With accelerating climate change, the impacts of hazards will compound and cascade'.

We corrected this (R2-m1)

Page 1, Line 15: Remove 'settings.

We corrected this (R2-m2)

Introduction -

Page 2, Line 39: More information on the 'four actors and two hazards' – who were the actors, were they all from one sector? Which hazards were addressed.

We corrected this:

"A recent case study on DAPP-MR with three sectors (agriculture, shipping, municipality) and two hazards (river floods and droughts) illustrated the difficulty in analyzing such multidimensional data, highlighting the need for better visualization tools to unravel complexity and support DRM \citep{Schlumberger.2024}.

Page 2 Line 53 - 55: This is more of a personal preference, but I don't see the point of the text from 'The paper is structured...' onwards and would delete everything that follows this in this section. It's a level of exposition I find unnecessary, Im reading the paper, therefore I know how its structured.

We agree and have removed it.

Methods -

Page 3, Line 63: Can you clarify the last sentence of this paragraph? I don't really understand what you mean here.

**We clarified:**

"These questions are translated into 'analysis operations' in using information visualization terminology to clarify the analysis goals and method."

Page 4, Line 79: 'Early adopters'. I'm assuming that this represents the field in general, not your interviewees. Perhaps clarify the language between the first and second sentence of this paragraph

We clarified language here:

"Multi-risk decision-making remains a relatively new and complex topic \citep{SakicTrogrlic.2024}, still largely situated in exploratory research and pilot initiatives. As a result, early adopters involved in pathways analysis come from diverse disciplines and administrative levels, motivated by...(i) understanding multi-risk interactions and system-wide effects, (ii) identifying sector-specific pathways with low costs and high (co-)benefits, and (iii) identifying system-wide low-regret pathways combinations."

Page 4, Line 81: What do you mean by 'low regret pathway'

We use different terminology to replace the term 'low-regret pathway'. See textual correction in the previous minor comment.

Page 4, Line 97: Remove 'Furthermore'.

We addressed this

Page 4, Line 98: Remove 'Furthermore, we used', replace with and.

We addressed this

Page 6, Line 125: Again, referring to a 'wide range of users', without identifying who these may be.

We changed into:

"When developing the interactive dashboard and integrating fit-for-purpose visualisations, we focused on two components: 1) designing information visualisations to complete the analysis operations and 2) creating an environment that serves different user types to gain additional insight into the concepts and purpose of the themes of analysis."

Page 7, Line 162: Unsure what you mean by 'bring their individual shortlisted pathways to one table'.

We clarified this as follows:

"In the last step of analysis, pathways from different sectoral actors are combined explore the interaction effects on the entire system."

Results -

Page 9: Are you missing a subheading? (3 Results)

We are not missing the subheadings.

Page 9, Line 190: What types of visual impairments are you referring to? Colour blindness?

We are referring to some color vision deficiencies. One candidate indicated they had Tritanomaly (difficulty to distinguish between blue and green) and one had difficulty perceiving red-green differences.

Page 10, Line 202: Every time you quote one of your interviewees you use a 'at the beginning of the phrase rather than a '- suspect that this is a LaTeX issue, please check.

Thanks. We corrected this.

Page 10: Change figure captions from left to a) and right to b) throughout the document?

We have updated the figures and references accordingly.

Page 12, Line 226: Can you clarify what you mean by evaluated here? As in you had a higher number of responses, or they found it easier to interpret.

Thanks for spotting this. We clarified: "It should be noted that the DMDU experts perceived PCP much more positively compared to the other expert groups, while the patterns were quite similar for SBC."

Page 12, Line 232 - 33: remove 'had a' change particular struggle to particularly struggled.

We changed that and checked the grammar and spelling of the whole manuscript again.

Page 12, Line 235 - 26: replace 'referred' with 'stated' and 'context' with 'contextual'.

We changed that and checked the grammar and spelling of the whole manuscript again.

Discussion -

Page 18, Line 351: I don't think that this reference is in the reference list.

Thanks for spotting this. We corrected the mistake.

Page 19, Line 363: There is a bad LaTeX link here for a reference.

This is a latex error. There should not be a third reference

Page 19, Section 4.3: Do you have any comment on how you would iterate your dashboard to take on board the feedback that you have received?

There are a couple of clear challenges that were mentioned by the feedback: e.g. color coding, additional hints on how to read, and potentially embedding short explanation video. The ability to interact with the visualisations could also be further improved, e.g. by allowing to make figures full screen, select the color palette themselves. Generally, the idea would be to make the prototype more generalizable by specifying a certain input format and inputs that allow for a modification of the dashboard and implementation in different case studies.

Page 20, Line 397: The statement 'This study also emphasises the value of interactive visualisations' appears contradictory – you have said in multiple times in earlier sections that your testers did not fully utilise these features, please clarify.

Some testers did not find (all) elements for interacting with the visualisations. While others did and appreciated their value. Our conclusion is that the interactions are quite crucial and helpful (to some) and - if better clarified what set of interactions are available - could be for more/all.

Page 20, Line 410: 'visualisation tools capable of clearly illustrating these complex interactions to help a decision maker' you haven't evidenced this by testing your approach with decision makers.

We don't claim that we offer evidence for a tool that does this, but rather that we think we found a promising starting point (see also the adjustments of the limitations in response to some of your major comments.

Page 20, Line 415: 'we assumed decision makers would first tackle sector specific risk strategies before incorporating multi-sectoral interactions' why? How do you evidence this? Does this genuinely reflect a 'decision makers' experience?

By Decision-maker we mean sectoral decision-makers. For example a farmer or an agricultural ministry. Just by how risk is (institutionally) siloed, there is a focus on the sectoral objectives first. This approach follows the same lines of argumentations as Schlumberger et al. (2022), proposing that pathways should first be developed for simplified sub-systems before integrating all information on a system level.

**Conclusion -**

Page 20, Line 427: Again, you have highlighted that participants valued the dashboards interactivity. But this seemingly contradicts the statements earlier on in the text, please consider how to evidence this more fully.

See previous comment.

**Figures -**

Figure 2 is labelled a - e, whereas later figures are labelled 'left / right'. Make labelling consistent, I think that labelling with letters is probably more appropriate.

**We addressed this.**

The matrices in figures 5, 6, 7, 8, 9 aren't intuitive to read, consider a different display? Perhaps bar charts?

We like this suggestion, and consequently replaced the heatmaps with bar charts.

Figure 4 – Check the x axis, category labels are overlapping. Legend boxes have been placed over data points – These need to be adjusted so that all the data points are clearly visible.

We adjusted Figure 4 accordingly.

Please note I have not completed a line by line check of the Appendices or references for expediency, so authors may wish to do this a final time.

We thank the reviewer for taking the time to provide very helpful feedback on the main text of the manuscript. We've gone through a final check of the appendices and references.